

# terrainbento 1.0: a Python package for multi-model analysis in long-term drainage basin evolution

Katherine R. Barnhart[1,2], Rachel C. Glade[2,3], Charles M. Shobe[1,2], and Gregory E. Tucker[1,2]

[1]Cooperative Institute for Research in Environmental Sciences, University of Colorado at Boulder, Boulder, CO, USA
[2]Department of Geological Sciences, University of Colorado at Boulder, Boulder, CO, USA
[3]Institute for Arctic and Alpine Research, University of Colorado at Boulder, Boulder, CO, USA

*Correspondence to:* Katherine Barnhart (katherine.barnhart@colorado.edu)

**Abstract.** Models of landscape evolution provide insight into the development of specific field areas, create testable predictions of landform development, demonstrate the consequences of current geomorphic process theory, and spark imagination through hypothetical scenarios. While the last four decades have brought the proliferation of many alternative formulations for the redistribution of mass by Earth surface processes, relatively few studies have systematically compared and tested these alternative equations. We present a new Python modeling package, terrainbento 1.0, that enables multi-model comparison, sensitivity analysis, and calibration of Earth surface process models. terrainbento provides a set of 28 model programs that implement alternative transport laws related to four model elements: hillslope processes, surface-water hydrology, erosion by flowing water, and material properties. The 28 model programs stem from 13 binary choices related to one of these four elements—for example, the use of linear or non-linear hillslope diffusion. terrainbento is an extensible framework: model base classes that treat the elements common to all models (such as input/output and boundary conditions) make it possible to create a new model without re-inventing these common methods. terrainbento is built on top of the Landlab framework, such that new Landlab components directly support the creation of new terrainbento models. terrainbento is fully documented, has 100% unit test coverage including numerical comparison with analytical solutions for process models, and continuous integration testing. We support future users and developers with introductory Jupyter notebooks and a template for creating new terrainbento model programs. In this paper, we describe the package structure, process model theory, and software implementation of terrainbento. Finally, we illustrate the utility of terrainbento with a benchmark example highlighting the differences in steady state topography between five different process models.





## 1 Introduction

Computational models of long-term drainage basin and landscape evolution have a wide spectrum of applications in geomorphology, ranging from addressing fundamental questions about how climatic

and tectonic processes shape topography, to performing engineering assessments of landform stability and potential for hazardous-waste containment (see, e.g., reviews by Coulthard, 2001; Pazzaglia, 2003; Martin and Church, 2004; Willgoose, 2005; Codilean et al., 2006; Bishop, 2007; Willgoose and Hancock, 2011; Pelletier, 2013; Temme et al., 2013; Valters, 2016). Although the basic principles of drainage basin evolution are reasonably well understood—such as the fundamental con-

cept that erosion is driven by gravitational and water-runoff processes, the latter of which depend strongly on surface gradient and water flow—there remains uncertainty concerning the appropriate forms of the governing transport laws for any particular set of materials and environmental conditions (Dietrich et al., 2003). This situation creates a need for comparative testing in order to gauge the overall performance of various model formulations, to identify knowledge gaps in areas where

models perform poorly, and to assess which transport laws are most appropriate for various types of environmental conditions, time scale, and spatial scale.

To date, there have been relatively few studies that have systematically compared and tested alternative transport laws, and those that do usually address only a single, quasi-one-dimensional landform element, such as the shape of an idealized hillslope (Roering, 2008; Doane et al., 2018), or

the longitudinal profile of a particular stream channel (Tomkin et al., 2003; van der Beek and Bishop, 2003; Valla et al., 2010; Attal et al., 2011; Hobley et al., 2011; Gran et al., 2013). Models that combine hillslope and channel processes—often referred to as Landscape Evolution Models (LEMs)—can simulate the formation of three-dimensional landforms that arise from interaction of multiple processes, and in principle comparative testing ought to be straightforward (Hancock et al.,

2010). Yet the algorithms behind these models commonly differ from one another in multiple ways, which makes one-to-one comparison difficult. For example, if two model codes differ simultaneously in their treatments of hydrology, sediment transport, and material properties, diagnosing any differences in their performance requires disentangling each of these effects. Often research questions focus on a combination of geological process and boundary conditions; classic examples include

morphologic dating of fault scarps and glacial moraines. In this way, boundary conditions become a core model component. To help address this challenge, it would be useful to have a software framework in which an investigator could alter one "process ingredient" or boundary condition at a time, and thereby conduct meaningful parameter studies, sensitivity analyses, calibrations, multi-model analyses, and comparisons with data.

terrainbento is a Python-language software product designed to help meet this need. terrainbento version 1.0 provides three resources for exploring alternative process models for landscape evolution. First, terrainbento 1.0 includes a collection of 28 distinct model programs for the long-term (order $10^4$–$10^6$ years) evolution of drainage basin topography; most of these models vary from a





simple "base" model in just one or two particular process descriptions. Second, terrainbento takes

advantage of Python class inheritance such that all common features of terrainbento models (such
as input/output, and the handling of boundary conditions) are provided in a generic "ErosionModel"
base class from which specific models are derived. This ErosionModel template enables modelers
to craft and apply their own model implementations without needing to re-invent the overarching
software framework or the necessary utility functions. terrainbento 1.0 builds on the Landlab Toolkit

(Hobley et al., 2017), using Landlab Components to represent individual hillslope, hydrologic, and
channel process components, and taking advantage of Landlab to handle common tasks such as input
and output management. Finally, model boundary conditions can have a profound impact on model
behavior. terrainbento has a set of extensible features called "boundary handlers" that can be used to
implement many common and complex boundary conditions.

Earth's landscapes are incredibly diverse, and the scientific questions that they pose are equally
extensive. No one model, or even a general framework like terrainbento, can hope to encompass all
of this diversity. terrainbento 1.0 was originally created to address landscape evolution in a humid-
temperate, soil-mantled, post-glacial environment with moderate relief (order $10^2$ m, on a horizontal
scale of order $10^4$ m) and relatively rapid erosion rates ($10^{-4}$ to $10^{-2}$ m/yr), over a time scale of order

$10^4$ years. The choices of algorithms and process laws among the constituent models reflect this
motivation. Nonetheless, through the model template, terrainbento provides a sufficiently generic
platform that it can be readily adapted to address a range of other scales and environments. This
paper presents and describes terrainbento version 1.0, including its basic structure, mathematical
underpinnings, software implementation, and the 28 constituent models.

## 80  2   General structure of a terrainbento model

A terrainbento model begins with a gridded representation of topography. By default, a regular raster
grid is used. Landlab's HexModelGrid type is also supported as an option. terrainbento version 1.0
does not support Landlab's irregular Delaunay-Voronoi grid type, but the framework could readily
be modified to accommodate it in the future.

The elevation, and regolith thickness if present, at each grid node evolves according to a specified
set of erosion and/or sediment transport laws, which vary from model to model. In this section, we
start by outlining the governing equations in a generic form. We then examine the software frame-
work that implements elements common to all terrainbento models. The subsequent section then
presents the collection of process laws and algorithms used to represent hillslope erosion, hydrol-

ogy, water erosion, and material properties. Section 4 then describes handling of boundary condi-
tions. The governing equations for all 28 models in terrainbento 1.0 are listed in Appendix B.





### 2.1 A note on terminology

The word "model" can have multiple meanings in scientific computing, and indeed in science generally. Here we will use the term *mathematical model* to mean a set of governing equations, which
in this case describe landscape evolution under a given set of assumed process dynamics, material properties, and boundary conditions. Under this definition, two mathematical models may have governing equations that are structurally quite similar, but which are nonetheless considered to be distinct models either because certain constants take on different values, or because a term is included in one version but not the other. For example, as described below, water erosion is commonly
treated as proportional to either hydraulic power or hydraulic stress. We consider these to be distinct mathematical models, despite the fact that the difference lies only in the choice of two exponent values in the governing equation.

Each *mathematical model* contains terms that represent individual processes (or closely related collections of processes), such as erosion by surface water flow. The mathematical representation
for an individual process will be referred to as a *process law* or *rate law*. By this definition, a *mathematical model* in terrainbento consists of a set of *process laws* embedded within an overall mass-conservation equation.

The term *numerical model* is used here to refer to a numerical algorithm that solves a particular mathematical model by marching forward in time from a given initial condition under given boundary conditions. The term *model program* will refer to a set of source-code files that performs the calculations needed to implement a *numerical model*. In some cases in terrainbento 1.0, a single *model program* can be configured to implement many *numerical models*, depending on its input parameters. One can consider alternative models that require only a different set of model program parameter values alternative *parametric models*, whilst models that require a different program are
different *structural models*. The combination of a model program plus the inputs that control this type of choice will be referred to as a *model configuration*.

### 2.2 Basic Ingredients and Governing Equation

Topography in a terrainbento model is represented as a two-dimensional field of elevation values, $\eta(x, y, t)$. The general governing equation describes the rate of change of $\eta$ as the sum of two terms:
one representing erosion (or deposition) of mass by water-driven processes, and one representing gravitational ("hillslope") processes:

$$\frac{\partial \eta}{\partial t} = -E_W - E_H \tag{1}$$

where $E_W$ is the rate of erosion (or deposition, if negative) by water-driven processes such as channelized flow, and $E_H$ is the rate for gravitationally driven processes such as soil creep and shallow
landsliding (the subscript $H$ stands for "hillslope," recognizing that gravitational processes will tend to be most important on hillslopes). Water erosion is assumed to depend on local slope gradient,


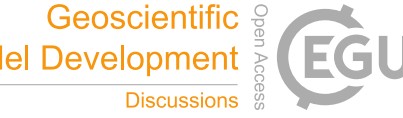


$S$, water discharge, $Q$ (which in many terrainbento models will be treated using drainage area as a surrogate, as discussed below), and material properties. Erosion or accumulation by gravitational processes is assumed to be a function of gradient, material properties, and (in some models) soil thickness.

While many terrainbento model programs only treat the evolution of the topographic surface $\eta$, two types of models treat a more complex set of layers. The first type adds an explicit mobile-regolith layer on top of bedrock. The second type treats two different bedrock lithologies in a user-determined spatial configuration.

### 2.2.1 Soil-tracking models

Several of terrainbento's models explicitly track a layer of regolith, defined here as unconsolidated and potentially mobile sediment, such as soil or alluvium (see Section 3.7.1). Here, for simplicity we will refer to this material as soil, keeping in mind that our operational definition is more general that the one commonly used by soil scientists. The land surface height is the sum of bedrock elevation, $\eta_b$, and soil thickness, $H$:

$$\eta = \eta_b + H. \tag{2}$$

Here too the term "bedrock" is used in its broadest possible sense, and may include for example cohesive sedimentary material such as glacial till. The time rate of change of soil thickness is the difference between the rate soil production and erosion,

$$\frac{\partial H}{\partial t} = P - E_{WHS}, \tag{3}$$

where $P$ is the rate of soil production from bedrock, and $E_{WHS}$ denotes the total rate of soil erosion (or accumulation, if negative) resulting from water-driven and gravity-driven transport processes. Similarly, the rate of change of bedrock surface height is the sum of soil production rate (scaled by any density contrast between rock and soil), and the rate of bedrock incision by running water, $E_{WR}$:

$$\frac{\partial \eta_b}{\partial t} = -\frac{\rho_s}{\rho_r} P - E_{WR}. \tag{4}$$

The rate of lowering of the bedrock surface is therefore the sum of the rate of rock-to-soil conversion and the rate of removal by water erosion.

### 2.2.2 Multi-lithology models

Nine terrainbento models allow for spatial juxtaposition of two different lithologies, $L_1$ and $L_2$. Layer $L_1$ is assumed to overlie $L_2$, but it may be absent (thickness zero) at any particular location. Let $\eta_{L2}(x, y, t)$ denote the elevation of the top of $L_2$, and $T_{L1}(x, y, t)$ represent the thickness of $L_1$. Then the land surface elevation (in the absence of an explicit soil layer) is given by:

$$\eta = \eta_{L2} + T_{L1}. \tag{5}$$





In models that honor both a soil layer and two different lithologies, the surface elevation is:

$$\eta = \eta_{L2} + T_{L1} + H, \tag{6}$$

in which case the height of the bedrock surface is

$$\eta_b = \eta_{L2} + T_{L1}. \tag{7}$$

Where the top layer exists, it lowers as a result of water erosion and (if soil is tracked) rock-to-soil conversion. This can be expressed mathematically as

$$\frac{\partial T_{L1}}{\partial t} = -\delta_L(E_W + P) \tag{8}$$

where $\delta_L$ is a spatially varying function equal to 1 where $L_1 > 0$, and 0 elsewhere (here $P$ is considered to be zero in non-soil-tracking models). The rate of change of elevation of the top of $L_2$ is given by

$$\frac{\partial \eta_{L2}}{\partial t} = -(1 - \delta_L)(E_W + P), \tag{9}$$

which simply means that the lower layer $L_2$ is vulnerable to erosion and weathering wherever the top layer is missing (for example, having been eroded through). Note that for reasons reflecting the original application of terrainbento, within the source code and input files the top layer is referred to as 'till' and the bottom layer as 'rock.' Note also that the BasicHySa model allows simultaneous water erosion of soil and rock, as discussed below.

## 3 Process formulations

Each model in the terrainbento 1.0 collection has four elements, reflecting the model's treatment of hillslope processes, surface-water hydrology, erosion by running water, and material properties. The possible formulations for each of these elements are constructed around a set of binary choices. Each choice represents a decision about how a particular element might be formulated. For example, the downhill soil transport rate could be represented as either a linear or nonlinear function of local topographic gradient, while the lithology could either be treated as being uniform, or divided into two distinct types as discussed in Section 2.2.2. The binary-choice design makes it possible to test the behavior of one alternative model element at a time. The binary options that form the basis for the terrainbento 1.0 constituent models are listed in Table 1. In Table 1, option B in each row usually represents a more sophisticated choice than option A: one that may bring more realism, but generally involves more parameters.

Each of terrainbento's models uses Landlab components to implement the numerical algorithms behind channel erosion, hillslope processes, and water-flow routing. The components used are briefly identified by name in the following descriptions of terrainbento model ingredients. The software





architecture that supports this component-based approach is then discussed further in Section 5. Further information about Landlab and its component-modeling capability is provided by Hobley et al. (2017).

### 3.1 Model domain options

terrainbento supports both regular raster grids and hexagonal grids. Raster grids may be initialized using an input Digital Elevation Model (DEM) or generated as a rectangular grid of user-specified dimensions and spacing with synthetic initial topography. Hexagonal grids may only be generated with synthetic topography. Many options for the creation of synthetic topography are available and are described in the User Manual.

### 3.2 Drainage area, flow direction, and flow accumulation

All terrainbento models calculate drainage area and surface water discharge using the Landlab FlowDirectors and FlowAccumulator. Flow direction algorithms presently supported in Landlab include: SteepestDescent/D4, D8, D$_\infty$, and Multiple Flow Direction. Water routing across closed depressions is optionally handled using a lake-fill algorithm implemented by the Landlab Depres-
205 sionFinderAndRouter component (the current version of which uses an implementation based on Tucker et al. (2001)). Once flow directions and surface water runoff are calculated, the contributing drainage area or surface water discharge at a given grid cell $i$ is calculated by adding up the area of all cells whose flow eventually passes through $i$, plus the area or discharge of $i$ itself using the Landlab FlowAccumulator component.



**Table 1.** Binary options for process formulations and boundary conditions.

| Category | Option A | Option B |
|---|---|---|
| Hillslope processes | linear transport law | non-linear transport law |
| Surface-water hydrology | deterministic | stochastic |
|  | uniform runoff | variable source area runoff |
|  | $\omega_c = 0$ | $\omega_c > 0$ |
|  | stream power | shear stress |
|  | constant $\omega_c$ | $\omega_c$ increases with incision depth |
|  | detachment-limited | sediment-tracking |
|  | uniform sediment* | fine vs. coarse* |
| Material properties | no separate soil layer | tracks soil layer $H(x,y,t)$ |
|  | homogeneous lithology | two lithologies |
| Paleoclimate | constant climate | time-varying $K$ |

* only applies to sediment-tracking model (see text).



**Table 2.** Summary of individual models in the terrainbento 1.0 collection.

| Model configuration | Element varied #1 | #2 | #3 |
|---|---|---|---|
| Basic | - | - | - |
| BasicTh | variable $\omega_c$ | - | - |
| BasicDd | $\omega_{ct} \propto$ incision depth | - | - |
| BasicHy | sediment-tracking channel erosion rule | - | - |
| BasicCh | nonlinear (cubic) hillslope soil transport | - | - |
| BasicSt | stochastic runoff generation | - | - |
| BasicVs | variable source area runoff generation | - | - |
| BasicSa | tracks soil/alluvium | - | - |
| BasicRt | tracks two lithologies | - | - |
| BasicCc | $K$ varies over time | - | - |
| BasicStTh | variable $\omega_c$ | stochastic runoff generation | - |
| BasicThVs | variable $\omega_c$ | variable source area runoff generation | - |
| BasicRtTh | variable $\omega_c$ | tracks two lithologies | - |
| BasicDdHy | $\omega_{ct} \propto$ incision depth | sediment-tracking channel erosion rule | - |
| BasicDdSt | $\omega_{ct} \propto$ incision depth | stochastic runoff generation | - |
| BasicDdVs | $\omega_{ct} \propto$ incision depth | variable source area runoff generation | - |
| BasicDdRt | $\omega_{ct} \propto$ incision depth | tracks two lithologies | - |
| BasicHySt | sediment-tracking channel erosion rule | stochastic runoff generation | - |
| BasicHyVs | sediment-tracking channel erosion rule | variable source area runoff generation | - |
| BasicHySa | sediment-tracking channel erosion rule | tracks soil/alluvium | - |
| BasicHyRt | sediment-tracking channel erosion rule | tracks two lithologies | - |
| BasicChSa | nonlinear (cubic) hillslope soil transport | tracks soil/alluvium | - |
| BasicChRt | nonlinear (cubic) hillslope soil transport | tracks two lithologies | - |
| BasicStVs | stochastic runoff generation | variable source area runoff generation | - |
| BasicSaVs | variable source area runoff generation | tracks soil/alluvium | - |
| BasicRtVs | variable source area runoff generation | tracks two lithologies | - |
| BasicRtSa | tracks soil/alluvium | tracks two lithologies | - |
| BasicChRtTh | nonlinear (cubic) hillslope soil transport | tracks two lithologies | variable $\omega_c$ |





### 3.3 Basic model

The simplest of the component models in terrainbento is known as the Basic model. Its governing
equation for land-surface elevation $\eta(x, y, t)$ is:

$$\frac{\partial \eta}{\partial t} = -KA^m S^n + D\nabla^2 \eta, \tag{10}$$

where $K$ is an erosion coefficient with dimensions of $[\text{L}^{1-2m}\text{T}^{-1}]$, $A$ is upstream contributing
drainage area, $S$ is gradient in the steepest down-slope direction, $m$ and $n$ are the area and slope ex-
ponents, respectively, and $D$ is a soil-creep coefficient with dimensions of $[\text{L}^2\text{T}^{-1}]$. Here the first term
on the right hand side represents channel and gully erosion, while the second term represents ero-
sion or deposition by gravitational sediment movement. An example of a landscape simulated using
the terrainbento Basic model, using the common choice $m = 1/2$ and $n = 1$, is shown in Figure 1a.
Each of the five panels in Figure 1 illustrates different terrainbento models constructed with the same
boundary conditions. The model runs that produced all of these example landscapes, including the
parameter dictionary that specifies the model run and slope area diagrams like that shown in Figure 2
can be found in the Jupyter Notebook tutorials on GitHub https://doi.org/10.5281/zenodo.1345788.

The second term on the right is the popular linear diffusion rule for hillslopes (Culling, 1963).
The first term on the right represents channel incision, and is based on the widely used stream-power
formulation (Howard et al., 1994; Whipple and Tucker, 1999), in which the long-term average rate
of channel downcutting is taken to be proportional to hydraulic power per unit bed area. A key
assumption behind the Basic model is that the erosion rate is limited by the capacity to detach
and remove material, rather than by along-stream variations in the capacity to transport sediment.
Drainage area appears as a surrogate for effective water discharge. Often the choice $m = 1/2$ is
made, reflecting the assumption that discharge per unit channel width scales as the square root of
drainage area. Similarly, $n$ is commonly taken to be unity, based on the derivation from stream
power. The examples presented below use the exponent values $m = 1/2$ and $n = 1$ unless otherwise
noted.

Rather than hard coding values for the water erosion drainage area exponent $m$ and slope exponent
$n$, terrainbento model programs permit these two exponents as parameters. Thus within terrainbento
1.0 the same model program can be configured to represent either a stream-power or shear-stress
representation of water erosion. Similarly, alternative models could be constructed by changing the
value of only $m$ or $n$ to reflect drainage area-channel width scaling (Snyder et al., 2003b; Wohl and
David, 2003) or different fluvial erosion processes (Whipple et al., 2000a). For example, Figure 1b
shows an example of an alternative *parametric model* which uses the Basic model with a value of
$m = 1/4$.

Although Equation 10 is rather simple, having just two parameters ($K$ and $D$), it represents a
formulation that has been widely used in geomorphic models (e.g., Miller and Slingerland, 2006;
Miller et al., 2007; Perron et al., 2009; Pelletier, 2010; Duvall and Tucker, 2015). The equations are




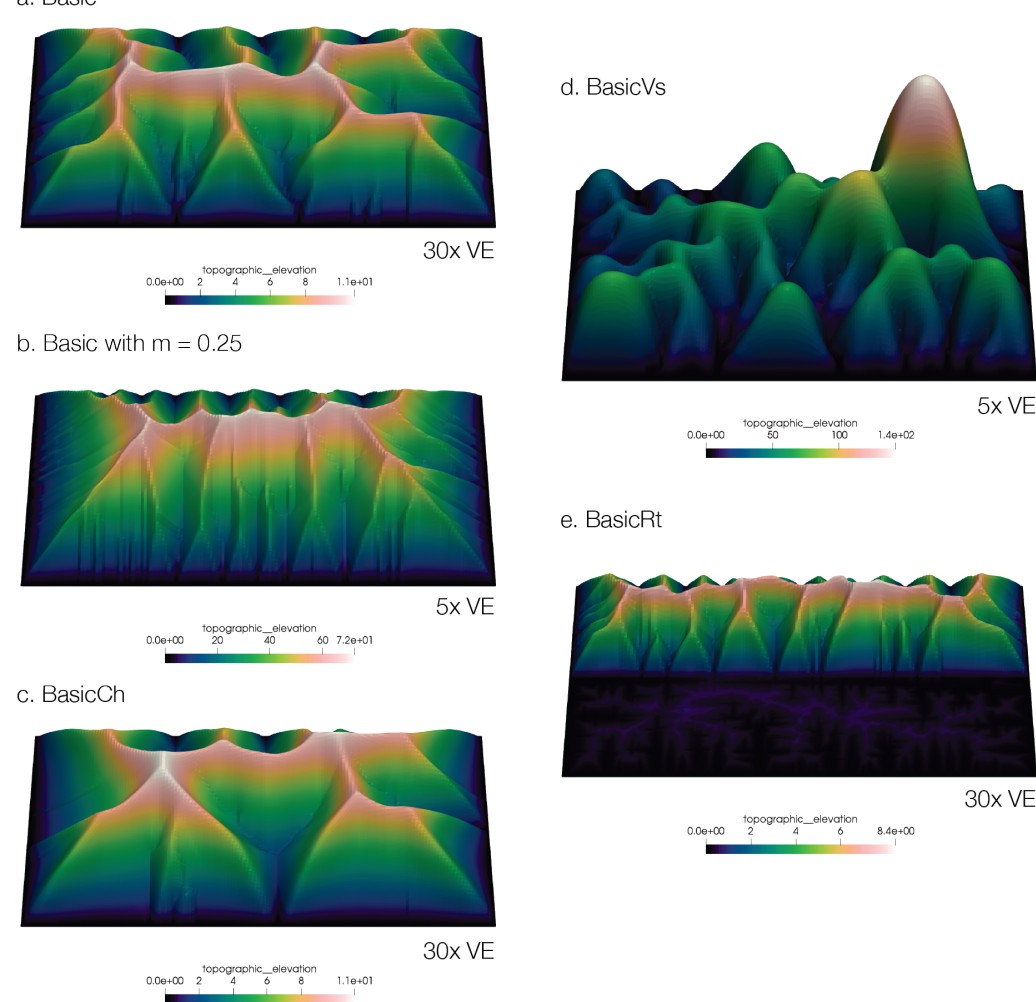

**Figure 1.** Three dimensional view of simulated topography illustrating the use of five different terrainbento models: a) model Basic, b) model BasicCh, c) model Basic with $m = 0.25$, d) model BasicVs, e) model BasicRt. Each landscape was initialized with the exact same random noise, has the same boundary conditions of the center core nodes uplifted relative to a fixed boundary, and was run to steady state (based on topographic change between 1000 year intervals exceeding 1 mm at no grid cell). Each model domain is 1x1.6 km, is represented at 10 m grid spacing, and has between 5 and 30x vertical exaggeration.

commonly solved numerically on a regular or irregular grid. The drainage area factor is normally evaluated using a downslope routing algorithm in which the water output from one grid cell is passed



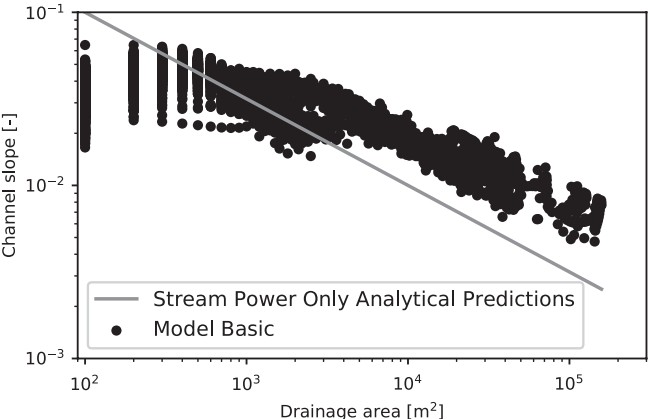

**Figure 2.** Slope area relationship for the example model in Figure 1

.

to one or more downhill neighboring cells (see, for example, review in Tucker and Hancock, 2010). Despite its simplicity, this two-parameter model has been shown to reproduce first-order proper-

250 ties of drainage basin topography, including dendritic drainage networks, concave-upward channel longitudinal profiles, and convex-upward hillslopes.

  One arrives at the terrainbento Basic model by choosing option A for each item in Table 1. In the following sub-sections, we review the various options that terrainbento offers for alternative treatment of hillslope processes, surface-water hydrology, channel incision, materials, and boundary

conditions.

### 3.4 Hillslope processes

To model hillslope evolution processes in a soil-mantled landscape, we use components of varying complexity that treat soil transport as a diffusion-like process in which sediment flux is governed by topographic gradient. terrainbento offers two alternative soil-flux rules with which to model the

260 downslope transport of soil and its dependence on topographic gradient: linear and nonlinear.

  In addition, as discussed previously, terrainbento also allows for the option of explicitly tracking a dynamic soil layer. This option is provided to address the possibility that soil may become thin enough to limit flux, and this limitation may in turn influence the rate and style of landscape evolution. Inclusion of a dynamic soil layer requires an equation for soil production from the underlying

lithology ($P$ in equation 3), and furthermore that the flux law be modified to account for the local soil thickness such that soil flux goes smoothly to zero as thickness declines.





### 3.4.1 Continuity law for soil creep

The simplest forms of the so-called "geomorphic diffusion" equation (Dietrich et al., 2003) assume transport-limited conditions, in which the production rate of soil is always much greater than the transport rate; thus, transport rate does not depend on soil availability or thickness. In this case, the hillslope term in the continuity equation (1) is:

$$E_H = \frac{1}{1-\phi}\nabla q_h \tag{11}$$

where $q_h$ is the hillslope soil volume flux per unit width, $\phi$ is the porosity of the soil, and the $\nabla$ operator represents differentiation in two horizontal directions ($\nabla = \partial/\partial x + \partial/\partial y$).

### 3.4.2 Linear creep law

A variety of formulae exist for the soil flux, $q_h$. The simplest and most common formula (Culling, 1963) treats the soil transport rate as a simple linear function of topographic gradient, using a transport efficiency constant, $D'$:

$$q_h = -D'\nabla\eta \tag{12}$$

where $\nabla\eta$ is the slope gradient. Using this flux rule with equation 11, the hillslope term in the continuity equation becomes:

$$E_H = -D\nabla^2\eta \tag{13}$$

where $D$, sometimes referred to as hillslope diffusivity, is equivalent to $D'/(1-\phi)$ and has dimensions of $L^2/T$. This simplest form of the evolution equation for soil creep on hillslopes results in convex-upward topography at steady state.

### 3.4.3 Nonlinear creep law

A more complex version of the creep law for soil-mantled slopes involves a nonlinear relationship between soil flux and topographic gradient. The nonlinear formulation captures accelerated creep and shallow landsliding as gradient approaches an effective angle of repose for loose granular material. Several nonlinear creep-transport laws have been suggested in the literature. The most popular of these is the Andrews-Bucknam equation (Andrews and Bucknam, 1987), which performs reasonably well when compared with experimental and field data (Roering et al., 1999, 2001; Roering, 2008). One problem with the Andrews-Bucknam law, however, is that the flux diverges when the slope gradient, $S$, equals the threshold gradient $S_c$, and is undefined for $S > S_c$. This property makes it challenging to incorporate into a landscape evolution model, where other processes may produce gradients equal to or greater than $S_c$. Some authors have addressed this problem with a modified form that avoids divergence at gradient $S = S_c$ (e.g., Carretier and Lucazeau, 2005).





terrainbento uses a truncated Taylor Series formulation for soil flux, which was derived by Ganti et al. (2012) for the Andrews-Bucknam law. The flux is given by

$$q_h = -DS \left[ 1 + \left( \frac{S}{S_c} \right)^2 + \left( \frac{S}{S_c} \right)^4 + ... \left( \frac{S}{S_c} \right)^{2(N-1)} \right] \tag{14}$$

where $S = -\nabla \eta$ is topographic gradient (positive downhill), $D$ is the transport efficiency factor, and $S_c$ is a critical gradient. The user specifies the number of terms $N$ to be used in the approximation. The nonlinear flux rule results in convex-up topography for shallow slopes, and transitions to linear hillslopes for steeper slopes. An example terrainbento simulation using the nonlinear creep law is shown in Figure 1c.

### 3.4.4 Linear depth-dependent creep law

For models that explicitly track a soil layer $H(x, y, t)$, one needs to modify the creep law to incorporate a relationship between flux, $q_h$, and local soil thickness. terrainbento uses an approach proposed by Johnstone and Hilley (2015), in which the flux decays exponentially as soil thickness approaches zero,

$$q_h = -D \left[ 1 - \exp \left( -\frac{H}{H_0} \right) \right] \nabla \eta, \tag{15}$$

where $H_0$ represents the soil thickness for which $q_h$ shrinks to $(1 - 1/e)$ of its maximum value for a given slope gradient. (Note that in the original formulation of Johnstone and Hilley (2015), $D$ is treated as the product of $H_0$ and a transport coefficient with dimensions of length per time; here we lump them together as $D$).

### 3.4.5 Nonlinear depth-dependent creep law

We can modify the nonlinear flux rule (equation 14) to accommodate soil, again assuming an exponential velocity distribution in the subsurface (Johnstone and Hilley, 2015):

$$q_h = -DS \left[ 1 - \exp \left( -\frac{H}{H_0} \right) \right] \left[ 1 + \left( \frac{S}{S_c} \right)^2 + \left( \frac{S}{S_c} \right)^4 + ... \left( \frac{S}{S_c} \right)^{2(N-1)} \right]. \tag{16}$$

This approach is somewhat similar to that used by Roering (2008) in a study that compared the predictions of a nonlinear, depth-dependent flux law with observed hillslope forms.

### 3.4.6 Soil production

Models that track a layer of soil must include an expression to specify the rate at which soil is produced from the underlying parent material. The most commonly applied formula, and the one used by terrainbento's soil-tracking models, treats the rate of soil production from the underlying



lithology as an inverse-exponential function of soil thickness (Ahnert, 1976; Heimsath et al., 1997; Small et al., 1999):

$$P = P_0 \exp(-H/H_s) \tag{17}$$

where $P_0$ is the maximum production rate (with dimensions of length per time), and $H_s$ is a depth-decay constant on the order of decimeters.

### 3.5 Hydrology

Treatments of surface-water hydrology in landscape evolution models are commonly quite straight-forward, reflecting the need for both simplicity and computational efficiency. Erosion formulae normally require specification of water discharge or (less commonly) depth. The most common parameterization is to use contributing drainage area, $A$, as a surrogate for surface-flow discharge, $Q$. This is the default option in terrainbento's models. Operationally, this means that the water-erosion law includes $A$ (see Section 3.6 below), and that the erosion law parameters embed information about climatic factors such as precipitation frequency and intensity, as well as material properties such as soil infiltration capacity (e.g., Tucker, 2004).

#### 3.5.1 Variable source-area hydrology

In vegetated, humid-temperate regions, storm runoff is commonly produced by the saturation-excess mechanism, in which rain falls on areas that have become saturated (Dunne and Black, 1970). Such areas tend to occur in locations with either gentle topography, large contributing area, or both. Because the source area for runoff generation is both limited in spatial extent and varies over time, the phenomenon has come to be known as variable source-area hydrology, or VSA for short. Previous modeling studies have suggested that VSA can impact long-term landform evolution, as steeper upland areas tend to experience less intense and/or less frequent erosion and sediment transport by runoff (Ijjasz-Vasquez et al., 1993; Tucker and Bras, 1998). For this reason, terrainbento 1.0 includes a set of models that provide a relatively simple treatment of VSA. This treatment is based on the approach of O'Loughlin (1986) and Dietrich et al. (1993), and is similar to the TOPMODEL concept of Beven and Kirkby (1979). Each element on the landscape is considered to have an upper permeable soil layer of thickness $H$ and saturated hydraulic conductivity $K_{sat}$. The soil layer is assumed to overlie relatively impermeable material. From Darcy's Law, the maximum shallow subsurface flow discharge when the soil is fully saturated is the product of conductivity, depth, and local hydraulic gradient, which is assumed to be equal to topographic gradient, $S$. The maximum subsurface discharge per unit contour width is therefore given by:

$$q_{ss} = K_{sat} H S = T S \tag{18}$$

where $T = K_{sat} H$ is the soil transmissivity. Next, we consider a recharge rate, $R$, which represents the average rate of water input per unit area (dimensions of length per time). The total unit discharge





is the product of recharge and drainage area per unit contour length, $a$:

$$q_{tot} = aR. \tag{19}$$

Using these two principles, the surface water unit discharge, $q$, is:

$$q = \begin{cases} 0 & \text{if } aR < TS \\ aR - TS & \text{otherwise.} \end{cases} \tag{20}$$

This threshold-based approach has been used, for example, in models that explore how hillslope
hydrology influences landform evolution (Ijjasz-Vasquez et al., 1993; Tucker and Bras, 1998). One
drawback, however, is that the use of mathematical thresholds in numerical models can compli-
cate the calibration process by creating "numerical daemons": sharp discontinuities in a model's
response surface (i.e., the $N_p$-dimensional surface that describes a particular model output quantity
as a function of its $N_p$ input parameters) (e.g., Kavetski and Kuczera, 2007; Hill et al., 2016). In this
particular case, we can create a smoothed version of (20) without any loss of realism, by positing
that within any given patch of land there is actually a distribution of effective recharge rates. The
simplest strictly positive probability distribution is an exponential function

$$p(R) = (1/R_m)e^{-R/R_m}, \tag{21}$$

where $p(R)$ is the probability density function of $R$, and $R_m$ is the mean recharge rate. The mean
surface-water unit discharge can then be found by integrating as follows:

$$\bar{q} = \int_{R_c}^{\infty} q(R)p(R)dR = aR_m e^{TS/R_m a}, \tag{22}$$

where $R_c = TS/a$ is the minimum recharge needed to produce surface runoff.

It is useful to re-cast this in terms of an effective contributing area, $A_{eff}$, defined as

$$A_{eff} = \frac{q\Delta x}{R_m} = Ae^{-T\Delta x S/R_m A} \tag{23}$$

where $\Delta x$ represents flow width (in a gridded digital elevation model, it would be natural to use
cell width). By this definition, the effective drainage area is always less than or equal to the actual
drainage area, reflecting the fact that some of the water runs through the shallow subsurface rather
than across the surface as overland (or channelized) flow. Where slope gradient is small or drainage
area is large, the effective area approaches the actual area. If the surface is flat ($S = 0$), the exponen-
tial factor equals unity and $A_{eff} = A$, reflecting the fact that no water can be conveyed by shallow
subsurface flow. Conversely, where $S$ is large and/or $A$ is small—as might be the case in steep head-
water areas—the effective drainage area becomes much smaller than the actual area, indicating that
most of the incoming water is traveling beneath the surface rather than contributing to overland flow.

A final step is to note that one can collapse the various factors in (23) into a single parameter,
$\alpha = T\Delta x/R_m$. This parameter has dimensions of length squared; we will refer to it henceforth as



the *saturation area scale*. A high value of $\alpha$ represents soils that have a large capacity to carry subsurface flow, relative to the recharge rate; a low value reflects a more limited subsurface flow capacity.

Seven of terrainbento's models implement variable source-area hydrology by using $A_{eff}$, as defined in (23), in place of drainage area, $A$ (Table 2). One of these (BasicSaVs) also explicitly tracks a soil layer, and the time- and space-varying thickness of this soil layer is used to calculate $T$ ($T = K_{sat}H(x,y,t)$) in this particular model. An eighth model (BasicStVs) also uses a stochastic treatment of precipitation; in this model, the randomly generated precipitation rate $p$ is used for $R_m$ in equation (23).

An example simulation with a terrainbento model (BasicVs) that includes a variable source-area component is shown in Figure 1d. The only difference in formulation between this example and the Basic model illustrated in Figure 1a is that BasicVs calculates channel erosion using effective drainage area, $A_{eff}$, as defined in equation (23), in place of total drainage area. The result is a drainage network bounded by steep, convex-upward ridges. These ridges are sufficiently steep that $A_{eff} \ll A$, so that their erosion is dominated by soil creep. The bases of the hills represent locations where water emerges from the shallow subsurface to become surface flow that feeds the channel network.

### 3.5.2 Stochastic precipitation and runoff

Many landscape evolution models use an *effective discharge* approach, in which a single value of precipitation or runoff (either given explicitly or embedded in a lumped rate coefficient) is used as a surrogate for the full range of runoff-producing events (e.g., Willgoose et al., 1991; Kooi and Beaumont, 1994; Tucker and Slingerland, 1997). This approach has the advantages of simplicity and computational efficiency, but also has limitations. For example, the appropriate effective discharge may vary in space and time (Huang and Niemann, 2006). One solution is to use a stochastic treatment of precipitation and/or discharge, in which events are drawn from a specified probability distribution (Tucker and Bras, 2000; Snyder et al., 2003a; Tucker, 2004; Lague et al., 2005).

In order to facilitate comparison between models with deterministic and stochastic treatments of water discharge, terrainbento 1.0 includes a set of six models that each implement two stochastic precipitation algorithms available in the PrecipitationDistribution Landlab component. The aim of these algorithms is not to reproduce individual storm events, but rather to capture a spectrum of runoff and stream-flow events of varying frequency and magnitude. The first of these two methods is a stochastic-in-time approach based on Tucker and Bras (2000). The second option uses deterministic time steps but stochastic precipitation intensity.

In the first option, a series of "storms" is generated based on a specified mean storm duration $T_r$, mean interstorm duration $T_b$, and mean storm depth $h_r$. The mean storm and interstorm durations are generated from exponential distributions, after Eagleson (1978). For each individual storm, the





mean storm depth is generated from a gamma distribution. The gamma shape parameter used to draw a random storm depth is equal to that specific storm's duration divided by the mean storm duration. The scale parameter is equal to the mean storm depth. The depth and duration of an individual storm are then used to calculate a rainfall intensity (Ivanov et al., 2007).

In the second option, in which the time-step duration is fixed, the frequency of occurrence of rainfall is described using an intermittency factor, $F$, which is defined as the fraction of time rain occurs rain, and a mean event precipitation rate, $p_d$.

Thus, the mean precipitation rate (averaged over wet and dry periods), $p_{ma}$, is given as

$$p_{ma} = F\, p_d\,. \tag{24}$$

The probability distribution of precipitation rate, $p$, is modeled using a stretched exponential survival function,

$$Pr(P > p) = \exp\left[-\left(\frac{p}{\lambda}\right)^c\right], \tag{25}$$

where $c$ is a shape parameter and $\lambda$ is a scale parameter. Use of the stretched exponential function is based on Rossi et al. (2016), who found that the function provides a good approximation for daily rainfall distributions in the continental US and Puerto Rico. While Rossi et al. (2016) found this distribution to be appropriate for mean *daily* rainfall, note that terrainbento is agnostic to the time units chose by a user. Wilson and Toumi (2005) argued that theoretical considerations suggest $c \approx 2/3$, while Rossi et al. (2016) found a mean value of $c = 0.74$ for weather stations in the continental US.

The shape parameter $\lambda$ associated with a mean daily precipitation rate $p_d$ and shape factor $c$ is given by

$$\lambda = \frac{p_d}{\Gamma(1 + \frac{1}{c})}\,, \tag{26}$$

where $\Gamma$ is the gamma function.

To describe the frequency-magnitude spectrum probabilistically in terrainbento's stochastic models, time is discretized into a series of steps of duration $\delta t/n_{ts}$, where $\delta t$ is the "global" time step. During each step, an "event" with precipitation rate $p$ is drawn at random from the cumulative distribution in equation (25). One of two approaches is then used to calculate the corresponding runoff rate, $r$. The first approach, which is the default used in five of the six stochastic models, assumes a mean soil infiltration capacity $I_m$. The rate of runoff is calculated as

$$r = p - I_m(1 - e^{-p/I_m})\,. \tag{27}$$

This formulation is a smoothed version of the simple threshold approach $r = \max(p - I_m, 0)$, which has been used in prior studies to represent infiltration-excess overland flow generation (e.g., Tucker and Bras, 2000). The smoothed version avoids the sharp discontinuity at $p = I_m$, and is arguably





more realistic as it honors natural variability in soil infiltration capacity. The runoff rate approaches zero when $p \ll I_m$, and approaches $p$ when $p \gg I_m$.

The second approach uses the variable source-area runoff generation model described in Section 3.5.1, using $p$ in place of recharge $R_m$. This approach is used only in model BasicStVs (Table 2).

### 3.6 Water erosion

Several different expressions have been proposed as models for long-term channel incision (and for erosion by surface water more generally). terrainbento 1.0 was originally designed to address erosion of cohesive sediments (including glacial till) and clastic sedimentary rocks with a relatively high fracture density, both of which are prone to erosion by hydraulic detachment of sediment grains and fracture-bounded fragments ("plucking"). This focus guided the choice of water-erosion laws

in terrainbento 1.0. Each terrainbento model uses one of two main types of erosion law: a simple area-slope detachment formula (sometimes referred to in the literature as the *stream power* family of erosion laws (e.g., Howard et al., 1994; Whipple and Tucker, 1999)), and an erosion formula that accounts for sediment discharge, particle entrainment from the bed, and particle deposition onto the bed. Within these two broad categories, terrainbento models express several variations in

form; for example, some include a threshold term, and in some of these the threshold increases with progressive incision depth. Each variation is presented and discussed in the sections below. Here, we start with a description of the simplest formulation, which serves as the default choice.

The area-slope (a.k.a., stream power) family of models derives from the assumption that the erosion rate, $E_W$, depends primarily on the hydraulic gradient, $S$, and the water discharge, $Q$,

$$E_W = k_1 Q^\mu S^\nu - \Omega_c \tag{28}$$

where $k_1$ is a coefficient that depends on material properties, channel geometry, and other factors, and $\Omega_c$ is a threshold below which no erosion occurs (in practice, the threshold is often assumed negligible, or its effects are taken to be subsumed in the exponents). The exponents $\mu$ and $\nu$ reflect the nature of the erosional processes; for example, Whipple et al. (2000a) argued that different val-

ues may be appropriate for abrasion-dominated and for plucking-dominated systems. The discharge exponent $\mu$ also embeds information about channel geometry. Often, drainage area $A$ is used as a surrogate for discharge. One limitation of equation (28) is that it does not allow for sediment deposition; for this reason, it is sometimes referred to as a *detachment-limited* law (a term first coined by Howard (1994)), reflecting the assumption that the rate of downcutting is limited by the rate at

which material can be detached and removed.

Despite the simplicity of equation (28), its various permutations have shown reasonable success when tested against field observations (Stock and Montgomery, 1999; Whipple et al., 2000b; Kirby and Whipple, 2001; Snyder et al., 2000; Lavé and Avouac, 2001; Tomkin et al., 2003; van der Beek and Bishhop, 2003; Duvall et al., 2004; Loget et al., 2006; Whittaker et al., 2007; Attal et al., 2008;





Yanites et al., 2010; Attal et al., 2011; Hobley et al., 2011; Gran et al., 2013). Landscape evolution models that use the generic stream-power approach are able to reproduce basic properties of erosional landscapes, such as dendritic channel networks with concave-upward longitudinal profiles (e.g., Howard, 1994; Whipple and Tucker, 1999; Tucker and Whipple, 2002).

  One of the most commonly used versions of equation (28) is obtained by making the following
assumptions: (1) effective discharge and channel width can be represented as a power functions of drainage area, and (2) the erosion threshold is negligible. Under these conditions, the erosion law becomes:

$$E_W = KA^m S^n, \tag{29}$$

where $K$ is a coefficient that includes information about precipitation and hydrology as well as
material properties and channel geometry. If one further assumes that (1) the rate of downcutting depends on stream power per unit surface area, (2) effective discharge is proportional to drainage area, and (3) channel width is proportional to the square root of discharge, then the exponent values become $m = 1/2$ and $n = 1$.

  The simplicity of equation (29)—it has only one parameter if $m$ and $n$ are assumed to be accu-
rate representations of process—together with its ability to reproduce common features of drainage basins and networks have led to its widespread use in landscape evolution studies, especially with the exponent values $m = 1/2$ and $n = 1$ (e.g., Duvall and Tucker, 2015). One might think of this particular configuration as the "model to beat": to justify a more complex formulation, one would ideally need to demonstrate that such a formulation performs distinctly better.

Equation (28), which we will refer to as the simple stream power law, forms the default choice for water erosion in terrainbento's model programs. By also providing models with alternative (often more complex) erosion laws to (28), terrainbento's model collection allows one both to compare the behavior of several different formulations, and to test their performance against data. In other words, terrainbento is designed to enable systematic, quantitative hypothesis testing among a collection of
different fluvial erosion laws. In the following sub-sections, we describe the variations and alternatives to simple unit stream power among the terrainbento 1.0 models. The complete governing equations for each of the terrainbento 1.0 models are given in Appendix A.

### 3.6.1 Erosion threshold

Bed-load sediment transport is well known to exhibit threshold-like behavior, in which the trans-
port rate is negligible until a certain minimum hydraulic tractive stress is reached, at which point significant transport begins. Similar behavior applies to the erosion of highly cohesive sediment (e.g., Julien, 1998), and presumably also to bedrock (though the values of the operative thresholds for bedrock are not known). For this reason, models of landscape or longitudinal channel profile evolution often include a threshold term below which no erosion takes place.



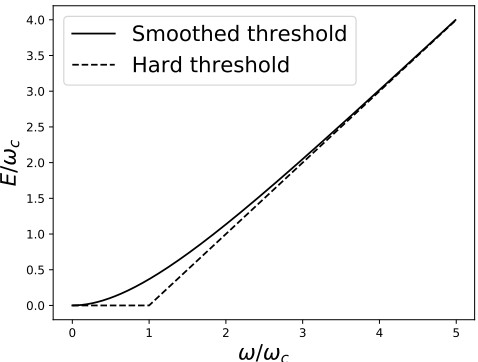

**Figure 3.** Illustration of the functional form of the smoothed-threshold erosion law (equation 30), compared with the more traditional hard-threshold formulation.

Several terrainbento models include a threshold in the water-erosion law. In order to promote mathematically smooth behavior, acknowledge evidence for distributions of transport thresholds (Kirchner et al., 1990; Wilcock and McArdell, 1997; McEwan and Heald, 2001), and avoid numerical daemons associated with threshold-type equations (e.g., Kavetski and Kuczera, 2007), the basic thresholded erosion law in terrainbento uses an exponential smoothing function following Shobe et al. (2017). terrainbento's thresholded erosion laws take the form:

$$E_W = \omega - \omega_c(1 - e^{-\omega/\omega_c}).\tag{30}$$

Here $\omega$ represents the erosion rate that would occur in the absence of a threshold, and is a function of slope gradient and either drainage area or discharge. For example, for those models that add a threshold term to the area-slope erosion in equation 29, $\omega$ is defined as

$$\omega = KA^mS^n.\tag{31}$$

The factor $\omega_c$ is a threshold with dimensions of length per time. The functional form of the smooth-threshold erosion function (equation 30) is illustrated in Figure 3. A constant threshold term is included in the water-erosion laws for five of terrainbento's constituent models (Table 2). Several others use a space- and time-varying threshold, as we describe next.

### 3.6.2 Incision depth-dependent erosion threshold

In a study of river incision into glacial deposits following ice recession in the US upper midwest, Gran et al. (2013) found evidence for an erosion threshold that increased with progressive incision depth. They attributed this to a downstream increase in median grain diameter resulting from enrichment of coarse gravel in bed material as the channel cuts through glacial deposits and the valley





widens. In comparing alternative long-profile evolution models with the observed profile, they found that the best match was achieved when the erosion threshold was allowed to increase linearly as a function of cumulative incision depth. Inspired by the findings of Gran et al. (2013), terrainbento 1.0 includes the option to allow the erosion threshold $\omega_{ct}$ to increase with erosion depth according to:

$$\omega_{ct}(x,y,t) = \max(\omega_c + bD_I(x,y,t), \omega_c) \tag{32}$$

where $D_I$ is the cumulative incision depth at location $(x,y)$ and time $t$, $\omega_c$ is the threshold when no incision has taken place yet, and $b$ (with dimensions of inverse time) sets the rate at which the threshold increases with progressive incision depth. As before, an exponential term is used to smooth the threshold, such that the water erosion rate approaches zero when $\omega \ll \omega_c$, and asymptotes to $\omega - \omega_c$ when $\omega \gg \omega_c$ (Figure 3). The max function is included to prevent the threshold from decreasing

in locations where hillslope processes produce net deposition (i.e., negative incision).

### 3.6.3 Shear-stress erosion law

Two important and commonly used measures of the erosional potential of stream flow are unit stream power and shear stress. The first represents the rate of energy dissipation per unit surface area, while the second represents the hydraulic traction force per unit area. Erosion rates in cohesive or rocky

material tend to correlate strongly with both quantities (e.g., Howard and Kerby, 1983; Whipple et al., 2000b), and both are widely used as the basis for long-term erosion laws. To support studies that compare and test these two approaches, terrainbento 1.0 allows one to configure the erosion law to represent bed shear stress rather than unit stream power. This is accomplished simply by changing the exponents on discharge (or drainage area) and channel gradient in equation (28). If one uses the

Manning equation to describe channel roughness and assumes that channel width is proportional to the square root of discharge, the applicable exponent values are $m = 3/5$ and $n = 7/10$ (Howard and Kerby, 1983; Howard, 1994). Use of the Darcy-Weisbach roughness law leads to a slightly different values, $m = 1/3$ and $n = 2/3$ (Tucker and Slingerland, 1997), which we use in the examples that accompany the terrainbento 1.0 documentation.

In terrainbento 1.0, the choice of exponent values is set using an input file, and so separate code is not needed to implement the shear-stress option. Nonetheless, we consider the stream-power and shear-stress formulations to form distinct parametric models.

### 3.6.4 Sediment-tracking entrainment-deposition hybrid model

The sediment-tracking model, following Davy and Lague (2009), computes changes in river bed

elevation resulting from competition between entrainment of bed material into the water column and deposition from the water column onto the bed. The governing equations, derived from a mass balance, state that changes in channel bed elevation $\eta$ over time are driven by bed material erosion





$E$ and bed material deposition $D_s$:

$$\frac{\partial \eta}{\partial t} = -E + \frac{D_s}{1-\phi} \tag{33}$$

where $E$ and $D_s$ are volume fluxes of bed material per unit bed area representing entrainment from
the bed and deposition onto the bed, respectively, and $\phi$ is the porosity of bed material. Note that
here our equation is different from Davy and Lague (2009) Equation 3 in that $1-\phi$ is not in the
denominator of $E$. This discrepancy is due to a difference in whether bulk or sediment density is
used to convert between mass and volume for $E$.

Equation 33 is coupled with conservation of sediment concentration in the water column of depth
$h$:

$$\frac{\partial (c_s h)}{\partial t} = E - D_s - \frac{\partial q_s}{\partial \hat{x}} \tag{34}$$

where $\hat{x}$ represents distance along the path of flow. The above states that sediment in the water
column involves a balance between erosion, deposition, and the streamwise spatial gradient in fluvial
sediment flux per unit width, $q_s$. Again following Davy and Lague (2009), we assume that the time
rate of change of sediment in the water column is negligible (as it is meant to represent an average
over time), so that

$$q_s = \int\limits_0^{\hat{x}} [E(\hat{x}) - D_s(\hat{x})] \, d\hat{x}. \tag{35}$$

In other words, the sediment flux at a particular downstream point $\hat{x}$ is the integral of all the erosion
minus deposition that has taken place upstream.

The erosion flux $E$ may be written in a number of ways, but in general depends on water discharge
$Q$ (or drainage area as a proxy), bed slope $S$, and some parameter or set of parameters describing
the erodability of the channel bed. As with other terrainbento models, we treat $m$ and $n$ as model
program parameters. The entrainment term may also include a threshold, and that threshold may be
constant or may vary with incision depth or with lithology.

Sediment deposition flux $D_s$ is a function of the concentration of sediment in the water column
$c_s$ and the effective settling velocity $V$ of the sediment particles. Adding that $c_s$ is the volumetric
sediment flux divided by the volumetric water flux, the deposition flux may be written:

$$D_s = V \frac{Q_s}{Q} \tag{36}$$

where $Q$ is volumetric water discharge and $Q_s$ is volumetric sediment discharge (equal to $q_s$ times
flow width). Importantly, $V$ is the net settling velocity after accounting for upward-directed turbu-
lence and sediment concentration gradients in the water column. Davy and Lague (2009) separate
the latter effects into a dimensionless parameter $d^*$ such that $D_s = d^* V \frac{Q_s}{Q}$, but here for simplicity
we combine both effects into an effective settling velocity $V$.



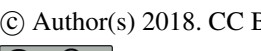

The entrainment-deposition model provides greater flexibility than detachment-limited models in that it can freely transition between detachment-limited and transport-limited behavior, depending on the relative importance of the erosion and deposition fluxes. If the deposition flux is negligible relative to the erosion flux, model behavior becomes detachment-limited. In the opposite case, the model becomes transport-limited. The entrainment-deposition model is therefore uniquely able to

treat landscapes that may exhibit both types of behavior at different points in space and time, at the cost of only a single extra parameter ($V$) relative to basic stream-power type models. For a full description of the entrainment-deposition model and its implications, see Davy and Lague (2009).

### 3.6.5    Entrainment-deposition hybrid model with fine sediment

In the entrainment-deposition approach proposed by Davy and Lague (2009), all material eroded

from the channel bed is included in sediment flux and deposition calculations. While this fully mass-conservative approach is a useful general case, it neglects the fact that clay- and silt-sized sediment may have such a low settling velocity as to be remain permanently suspended until and unless they enter a body of standing water. A simple modification to the entrainment-deposition model allows for treatment of a scenario in which the finest fraction of eroded sediment is in permanently from

the erosional landscape upon entrainment. In the general case, the change in $Q_s$ along the river is written:

$$\frac{\mathrm{d}Q_s}{\mathrm{d}x} = E\mathrm{dx}_f - D_s\mathrm{dx}_f. \tag{37}$$

where $dx$ is the width of flow. To account for permanently suspendable fine sediment, represented as a fraction of total bed sediment $F_f$, we simply exclude the fine sediment from the sediment flux

and write:

$$\frac{\mathrm{d}Q_s}{\mathrm{d}x} = (1 - F_f)\,E\mathrm{dx}_f - D_s\mathrm{dx}_f \tag{38}$$

such that the material incorporated into the sediment flux is reduced in proportion to the amount of fine sediment on the bed. This approach is simple and efficient, but would likely be limited in settings with very high proportions of fine sediment, as large concentrations of even very fine grains

in the water column may inhibit further sediment entrainment (Davy and Lague, 2009).

### 3.6.6    Entrainment-deposition model with bedrock and alluvium

One weakness of the erosion-deposition model described above is its limitation to a single type of bed material. For example, one can configure the parameters to represent erodible material such as loose sediment, or resistant material such as indurated bedrock, but not both at once. This limitation

means that the basic form of the entrainment-deposition model cannot honor the reality that many bedrock-incising rivers are blanketed by alluvium, nor can it be used to assess the relative contributions of sediment entrainment and bedrock erosion to channel morphology and sediment flux. One



potential solution is to use the erosion-deposition model in conjunction with a substrate layering system (i.e., a layer of sediment overlying bedrock), in which each layer is defined by its own erod-
ability factor and erosion threshold (e.g., Gasparini et al., 2004; Carretier et al., 2016). However, such an approach does not allow the simultaneous erosion of sediment and bedrock, which can occur in real rivers when the alluvial cover is spatially discontinuous and/or intermittent in time. Some recent modeling approaches allow a smooth transition between alluviated and bare-bedrock beds, and simultaneous evolution of the sediment and bedrock surfaces (Lague, 2010; Zhang et al., 2015; Shobe
et al., 2017). Lague (2010) tracked sediment thickness and allowed progressively more bedrock erosion as sediment thickness $H$ declined relative to median grain size $D_{50}$. He tested both exponential and linear models for the relationship between bedrock exposure and the ratio $H/D_{50}$. Zhang et al. (2015) compared sediment thickness to a statistical description of the macro-scale bedrock roughness to determine the probability of bedrock being exposed. The probability of bedrock exposure
increased with declining sediment thickness and increasing bedrock surface roughness.

In terrainbento we use the Landlab Component developed by Shobe et al. (2017), which expresses the Stream Power with Alluvium Conservation and Entrainment [SPACE] model. They used an exponential expression describing increases in bedrock exposure as sediment thickness declines relative to bedrock surface roughness. The SPACE model tracks topographic elevation $\eta$ as well as
bedrock surface elevation $\eta_b$ and sediment thickness $H$, such that

$$\frac{\partial \eta}{\partial t} = \frac{\partial \eta_b}{\partial t} + \frac{\partial H}{\partial t}. \tag{39}$$

Changes in sediment thickness are treated identically to the erosion-deposition model (equation 33), and changes in bedrock height are driven by bedrock erosion $E_r$ (there is no deposition of bedrock):

$$\frac{\partial \eta_r}{\partial t} = -E_r. \tag{40}$$

Erosion and deposition of sediment are computed using the same approach as used in the more basic entrainment-deposition model, with the addition of a factor that limits the rate of sediment entrainment, $E_s$, as sediment availability declines:

$$E_s = K_s A^m S^n \left(1 - e^{-H/H_*}\right). \tag{41}$$

where $K_s$ is an entrainment coefficient for alluvium. Here $H_*$ is the bedrock surface roughness length scale. Large $H_*$ corresponds to a rough bedrock surface and vice versa.

The SPACE model includes a similar formulation for the bedrock, where bedrock erosion becomes more efficient as sediment thickness declines:

$$E_r = K_r A^m S^n e^{-H/H_*}. \tag{42}$$





Here, $r$ subscripts denote bedrock parameters. Adding bedrock erosion to the entrainment-deposition model requires that eroded bedrock material be added to sediment flux calculations:

$$\frac{\mathrm{d}Q_s}{\mathrm{d}A(\hat{x})} = E_s + (1 - F_f)E_r - D_s, \tag{43}$$

where $A(\hat{x})$ represents drainage area, which increases as a function of streamwise distance $\hat{x}$. The factor $F_f$ indicates the proportion of the bedrock that is made up of fine sediment that goes into
permanent suspension when entrained, and is no longer included in model calculations. $Q_s$ therefore only includes grains not considered "fine."

As demonstrated by Shobe et al. (2017), the SPACE model is capable of transitioning between detachment-limited and transport-limited behavior. In a further advance over basic entrainment-deposition models, SPACE can model bare-bedrock channels, fully alluvial channels, and mixed
bedrock-alluvial channels, allowing the transition between these states to be set by sediment flux and erosive power. SPACE enables modeling of channels that may alternate between bedrock, bedrock-alluvial, and alluvial states in response to changing tectonic forcing, climate, or sediment supply conditions. For a full derivation and discussion of the SPACE model, as well as a development of steady-state analytical solutions, see Shobe et al. (2017).

### 3.6.7 How the alternative hydrology models influence terrainbento's erosion laws

For those models that use variable-source area hydrology, the drainage area factor in the water-erosion law is replaced by effective drainage area, $A_{eff}$, as defined by equation (23). Models that use stochastic hydrology replace $A$ with $Q = rA$, using $r$ as defined in equation (27).

One model, BasicStVs, combines stochastic runoff generation with variable source-area hydrol-
ogy. With this model, as in the variable-source model more generally, the capacity to carry subsurface discharge is defined as

$$Q_{ss} = TS\Delta x, \tag{44}$$

where as before $T$ is transmissivity, $S$ is surface gradient, and $\Delta x$ is flow width. Assuming interception loss and leakage to deeper groundwater are negligible, the total discharge produced by a storm
event with rainfall rate $p$ is

$$Q_{tot} = pA. \tag{45}$$

The surface discharge, $Q$, should then be the difference between these two quantities, or zero if $Q_{ss} > Q_{tot}$. However, a simple "either-or" differencing formulation is somewhat unrealistic (given small-scale natural variability in $T$), and if implemented numerically would risk creating numeri-
cal daemons in the model's response surface. To avoid these issues, the BasicStVs model uses the exponentially smoothed formula

$$Q = Q_{tot} - Q_{ss}[1 - \exp(-Q_{tot}/Q_{ss})], \tag{46}$$





so that $Q \to 0$ when $Q_{tot} \ll Q_{ss}$, and $Q \to Q_{tot}$ when $Q_{tot} \gg Q_{ss}$. The form of this equation is similar to that of the smooth-threshold erosion law illustrated in Figure 3. Substituting the definitions of $Q_{tot}$ and $Q_{ss}$ above,

$$Q = pA - TS\Delta x[1 - \exp(-pA/TS\Delta x)]. \tag{47}$$

The precipitation rate calculated for each stochastic event is used to calculate $Q$, which is then used as the discharge factor in the erosion law $E_W = K_q Q^m S^n$.

### 3.7 Material properties

#### 3.7.1 Soil and alluvium

One of the binary options listed in Table 1 is the ability to track explicitly a dynamic soil layer. Models that use this option implement the depth-dependent form of the applicable soil-creep law (i.e., either the linear or nonlinear form).

When the dynamic-soil option is used in combination with a sediment-tracking entrainment-deposition erosion law (model BasicHySa), the SPACE model described above is used in place of the simpler (single-material-type) entrainment-deposition law. In all other cases, the use of a dynamic soil layer does not directly influence the water-erosion law.

When dynamic soil is combined with variable source-area hydrology (model BasicSaVs), the actual soil thickness at each point $H(x, y, t)$ is used to calculate transmissivity.

#### 3.7.2 Multiple lithologies

With two-lithology models, the material-dependent parameters in the water-erosion equation, including the coefficient ($K$, $K_{ss}$, or $K_q$) and, if applicable, the threshold ($\omega_c$), vary in space and time as a function of the local surface elevation, $\eta$, in relation to the elevation of the contact between lithologies 1 and 2, $\eta_C(x, y)$. If $\eta > \eta_C$, lithology 1 is exposed at the surface; otherwise, the surface unit is lithology 2.

To acknowledge that lithological contacts are not razor thin and to preserve smoothness in the numerical solution, we allow there to be a finite "contact zone" within which the two lithologies are both considered to influence the material erodability. One might imagine this zone as representing a gradational transition from one unit to another, or alternatively an uneven contact surface. We define a weight factor $w$ that defines the relative influence of each of the two lithologies:

$$w(x, y, t) = \frac{1}{1 + \exp\left(-\frac{(\eta - \eta_C)}{W_c}\right)}. \tag{48}$$

Here, $w$ represents the influence of lithology 1, and $1 - w$ describes the influence of lithology 2. At each location, the channel erosion rate coefficient is calculated by applying this weight factor. For example, in model BasicRt, which uses the simple unit stream power formula, the rate coefficient $K$





is calculated as

$$K(\eta, \eta_C) = wK_1 + (1-w)K_2 \tag{49}$$

where $K_1$ and $K_2$ are the rate coefficients associated with each lithology, and $W_c$ is the contact-zone width.

### 3.8 Variable climate

As a simpler representation of variable climate than available in the PrecipChanger described in Section 4.2, one model (BasicCc) provides the ability to change parameter $K$ linearly through time. The representation of change is as follows. At the beginning of a model run, $K$ is assumed to be larger or smaller than its final value ($K_0$) by a factor $f$; if $f > 1$, $K$ starts out larger than $K_0$ (representing a more erosive climate) and declines through time, and conversely if $f < 1$. $K$ stops changing after

a time period $T_s$, whereupon it assumes its final value $K_0$. Mathematically, this linear variation in $K$ is

$$K(t) = \begin{cases} \kappa t + fK_0, & \text{when } t < T_s, \\ K_0 & \text{otherwise.} \end{cases} \tag{50}$$

where $\kappa = (1-f)K_0/T_s$ is the rate of change.

### 3.9 Pairwise process combinations

As noted earlier, the various process-model options described above can be arranged into a set of 11 binary choices (Table 1). terrainbento 1.0 is designed to support experimentation and hypothesis testing among these (and other) alternative formulations. The number of possible unique combinations among this set of 11 options is unwieldy ($2^{11}$, though some are not physically sensible). In creating the individual terrainbento model configurations, we used an approach that focuses on single and

pairwise variations on the Basic (simplest) model, which is the first entry in Table 2. The next 11 entries are models or model configurations that differ from Basic in just one element. The remaining entries represent pairwise combinations. Not all possible pairwise combinations are included. Instead, the pairwise process combinations selected represent those for which we thought there might be nonlinear interactions between the two process elements—in other words, those combinations

where we expected the whole to be greater (or less) than the sum of the parts. An example of such a nonlinear interaction that has been explored in the literature is temporal variability in water discharge in a river system where the erosion process is strongly thresholded (Tucker and Bras, 2000; Snyder et al., 2003a; Lague et al., 2005; DiBiase and Whipple, 2011). This particular combination is represented in terrainbento by model BasicStTh.

The particular list of model choices in Table 2 is not meant to be exhaustive. The terrainbento software was designed to be easily extensible as needed for any given application, so that for example



if a researcher wishes to explore combinations that are not included in the present collection of models, or to add a new process formulation, he or she can do so with relative ease. In the next section, we describe how the software is designed to promote extensibility.

## 4 Boundary conditions

Just as process representation influences model results, so do model boundary conditions. Representing boundary conditions in a component-like fashion permits systematic and reproducible changes in boundary conditions through either boundary condition component choice or parameter choice. To support alternative boundary conditions, terrainbento 1.0 includes five boundary condition handler classes. These boundary condition handlers are similar in construction to Landlab components: they are Python objects, they must have an __init__ method that takes as a first argument a Landlab model grid, and a run_one_step method that takes as its only argument the timestep duration dt. Four of these classes are called "Baselevel Handlers," reflecting that they modify the elevations on the boundaries of the modeled terrain. The final class is the PrecipChanger, a boundary condition handler that either modifies precipitation statistics or the value of $K$, $K_r$, $K_s$, $K_1$, or $K_2$, depending on the model.

### 4.1 Baselevel Handlers

Each of the four baselevel handlers modifies the elevations of specific grid nodes. Before describing these baselevel handlers it is worth reviewing the boundary condition types available to Landlab model grid nodes (Hobley et al., 2017). A boundary node can be open (with either a fixed value or fixed gradient), looped, or closed. A boundary node need not live on the edge of a rectangular grid—for example, many nodes may be closed boundary nodes if the model domain is a single watershed. All nodes that are not boundary nodes are called "core nodes."

The four baselevel handlers were designed to capture the most common cases for boundary conditions in Earth surface processes modeling. The SingleNodeBaselevelHandler controls the elevation of a single open, fixed-value boundary node, meant to represent a watershed outlet. The outlet lowering rate is specified either as a constant or through a time or through a user-supplied text file that specifies the elevation change through time. The NotCoreNodeBaselevelHandler moves either the core nodes or the not-core nodes at a constant rate through time or based on a text file. The CaptureNodeBaselevelHandler was designed to simulate drainage basin capture by a basin external to the model domain. It changes the boundary condition status of a single node from closed to fixed-value open at a user defined time and lowers its elevation.

The final baselevel handler is the GenericFunctionBaselevelHandler. It is similar to the NotCoreBaselevelHandler in that it either moves the core nodes or the not-core nodes. However, instead of taking a constant rate or time-elevation pattern as input, it requires that a user define a function





of two arguments that returns an at-node field of uplift rate. The two required arguments are the model grid and the model integration time. As the model grid contains attributes `x_of_node` and `y_of_node`, this boundary condition handler thus permits a user to define the relative uplift rate as any function of space and time.

### 4.2 PrecipChanger

The final boundary condition handler was designed to implement the impacts of changing climate on the precipitation distribution and, by extension, the erodability of material by water. For models with stochastic precipitation and uniform time steps, this method modifies the intermittency factor and the mean rainfall rate, whereas for models with an effective discharge it modifies the erodability by water. This boundary condition handler does not presently support stochastic precipitation with stochastic event durations.

For "St" models that explicitly represent the intermittency factor $F$ and mean rainfall rate $p_d$ (Section 3.5.2), the PrecipChanger must only modify those parameters. For models with effective discharge, deriving a relation between $K$ (or $K_r$, $K_s$, $K_1$, or $K_2$), $p_d$, and $F$ requires defining an underlying hydrology model and tracing how variations in precipitation model parameters influence the long term erosion rate. We start by noting that drainage area serves as a surrogate for discharge, $Q$. We can therefore write an instantaneous version of the erosion law in the Basic models as

$$E_i = K_q Q^m S^n. \tag{51}$$

This formulation represents the erosion rate during a particular daily event with daily-average discharge $Q$, as opposed to the long-term average rate of erosion, $E$. We next assume that discharge is the product of runoff rate, $r$, and drainage area:

$$Q = rA. \tag{52}$$

Combining these we can write

$$E_i = K_q r^m A^m S^n. \tag{53}$$

This equation establishes the dependence of short-term erosion rate on catchment-average runoff rate, $r$.

Next we need to relate runoff rate to precipitation rate. A common method is to acknowledge that there exists a soil infiltration capacity, $I_c$, such that when $p < I_c$, no runoff occurs, and when $p > I_c$,

$$r = p - I_c. \tag{54}$$

An advantage of this simple approach is that $I_c$ can be measured directly or inferred from streamflow records.



To relate short-term ("instantaneous") erosion rate to the long-term average, one can first integrate the erosion rate over the full probability distribution of daily precipitation intensity. This operation

yields the average erosion rate produced on wet days. To convert this into an average that includes dry days, we simply multiply the integral by the wet-day fraction $F$. Thus, the long-term erosion rate by water can be expressed as:

$$E = F \int_{I_c}^{\infty} K_q (p - I_c)^m A^m S^n f(p) dp, \tag{55}$$

where $f(p)$ is the probability density function (PDF) of daily precipitation intensity. By equating the

above definition of long-term erosion $E$ with the simpler definition in equation (51), we can solve for the effective erosion coefficient, $K$:

$$K = F K_q \int_{I_c}^{\infty} (p - I_c)^m f(p) dp. \tag{56}$$

In this case, what is of interest is the *change* in $K$ given some change in precipitation frequency distribution $f(p)$. Suppose we have an original value of the effective erodability coefficient, $K_0$, and

an original precipitation distribution, $f_0(p)$. Given a future change to a new precipitation distribution $f(p)$, we wish to know what is the ratio of the new effective erodability coefficient $K$ to its original value. Using the definition of $K$ above, the ratio of old to new coefficient is:

$$\frac{K}{K_0} = \frac{\int_{I_c}^{\infty} (p - I_c)^m f(p) dp}{\int_{I_c}^{\infty} (p - I_c)^m f_0(p) dp} \tag{57}$$

Thus, if we know the original and new precipitation distributions, we can determine the resulting

change in $K$.

We use a Weibull distribution for the precipitation intensity PDF (Rossi et al., 2016),

$$f(p) = \frac{c}{\lambda} \left( \frac{p}{\lambda} \right)^{(c-1)} e^{-(p/\lambda)^c} \tag{58}$$

where $\lambda$ is the distribution scale factor. Its relationship with $p_d$ is defined as

$$p_d = \lambda \Gamma (1 + 1/c). \tag{59}$$

The above definition can be substituted into the integrals in equation (57). We are not aware of a closed-form solution to the resulting integrals. Therefore, the erosion models used for projection apply a numerical integration to convert the input values of $F$, $c$, and $p_d$ (the last of which can change over time) into a corresponding new value of $K$.

## 5 Software implementation

### 5.1 Overview

In creating a software product that manifests not one but rather dozens of potential model configurations, efficiency and reuse are key design considerations. To meet this goal, terrainbento 1.0 uses





an object-oriented approach to its high-level design. Each terrainbento model is implemented as a
Python class. The class that implements any particular terrainbento model inherits from a common
base class called ErosionModel. Here we describe the main functions of the base class, the typical
structure of the derived class, and the use of a driver program to configure and execute a terrainbento
model.

### 5.2   terrainbento base classes

terrainbento contains three base classes to minimize duplicate code and maximize extensibility of
the model framework. The first of these, ErosionModel, handles common model instantiation, run,
output creation, and model finalization methods. These include creating the model grid, reading
initial topography from a file, creating synthetic topography, calculating elevation change, writing
netCDF and xarray datasets of model output, and interfacing with the boundary condition handlers.
All models except the "St" and "Rt" series inherit directly from the ErosionModel base class.

The stochastic and two-lithology models each have a sufficient number of specialized methods to
justify having their own base classes, which are the StochasticErosionModel and TwoLithologyEro-
sionModel, respectively. Both of these inherit from ErosionModel. The StochasticErosionModel
handles setting up the stochastic rain generator; calculating precipitation, runoff, and water erosion;
and keeping records of storm sequences. The TwoLithologyErosionModel handles setting up the
lithology contact elevation and updating any fields that depend on the depth to the contact.

### 5.3   Basic Model Interface

The Community Surface Dynamics Modeling System (CSDMS) has promoted use of an interface
standard known as the Basic Model Interface (BMI) for geoscientific numerical models (Peckham
et al., 2013). Although terrainbento does not yet fully implement a BMI, its model-control functions
follow the conventions used by the Landlab Toolkit, which themselves have a close parallel with
the main BMI model-control functions. The terrainbento `initialize` method is fully compatible
with the BMI method of the same name, which takes as an argument a string containing the name of
a parameter-input file (terrainbento's version can alternatively accept a Python dictionary containing
parameter name-value pairs). The terrainbento `run_one_step` method serves the same function
as BMI's `update`, but accepts step size as an argument. terrainbento's `run_for` is similar to BMI's
`update_until` (the former takes a duration whereas the latter takes an absolute time).



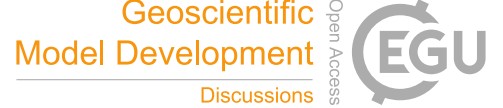

**Table 3.** Public base class methods.

| Name | Purpose |
|---|---|
| `__init__` | Initialize model |
| `run_one_step` | Execute one time-step of duration $dt$ [1] |
| `run_for` | Call `run_one_step` repeatedly to execute model for given total duration |
| `run` | Execute complete model run, pausing periodically to write output |
| `write_output` | Write output at current model time to netCDF file |
| `calculate_cumulative_change` | Calculate cumulative node-by-node changes in elevation |
| `finalize__run_one_step` | Update model time and boundary conditions |
| `finalize` | Clean up prior to ending execution[1] |
| `to_xarray_dataset` | Convert all previously written netCDF files to a single xarray dataset |
| `save_to_xarray_dataset` | Save a model run xarray dataset to a netCDF file |
| `remove_output_netcdfs` | Cleanup single timestep netCDF files |

[1] empty function intended to be overridden by child class.

### 5.4 Derived classes and use of Landlab Components

Two features make the process of writing a new model program in terrainbento relatively fast and efficient: the ability to inherit functionality from the terrainbenot base classes, and the use of Process

Components in the Landlab Toolkit to handle individual process laws. Having already discussed the base class, it is useful to say a few words about Landlab. The Landlab Toolkit is a Python-language software library designed to support efficient creation, exploration, and modification of two-dimensional numerical models of earth-surface processes (Hobley et al., 2017). Landlab accomplishes this by using a CSDMS-inspired plug-and-play method, in which the functionality needed

for a numerical implementation of a single process is encapsulated in a standard-format *Process Component*. Process Components are implemented as Python classes. Landlab also uses an object-oriented approach to grid creation and management, so that a simulation grid is encapsulated as a Python object. Components normally interact with a Grid object, and share *fields* (arrays) of grid-linked data by creating and attaching the necessary fields to a common grid. More information about

Landlab can be found in Hobley et al. (2017).

terrainbento uses Landlab Components to implement its process laws. Each terrainbento model program is implemented as a class that derives from the ErosionModel base class. The model program's `__init__` method handles parameter retrieval, and instantiates the necessary Landlab Components. The model program's `run_one_step` method then advances each component in turn,

normally by calling the component-level `run_one_step`. In addition to the definition of the model class, each terrainbento model program includes a short `main` function that allows the model program to be run in a stand-alone fashion (as opposed to being instantiated and run from an outside





```
# Example inputs for terrainbento model Basic

# parameters required by all models
dt: 10 # years
output_interval: 1e4 # years
run_duration: 1e7 # years

# parameters that specify the details of the model grid
number_of_node_rows: 100
number_of_node_columns: 160
node_spacing: 10.0 # meters
add_random_noise: True
initial_noise_std: 1.
random_seed: 4897 # initialize with reproducible random noise

# parameters that control geomorphic processes
water_erodability: 0.001 # years^-1
m_sp: 0.5 # unitless
n_sp: 1.0 # unitless
regolith_transport_parameter: 0.2 # meters^2/year
```

**Figure 4.** Example of a terrainbento input file.

script, which can also be done). This simple design allows the main model program files to be quite short, often with between 100 and 300 lines, of which only 20-50 are "true" lines of code and the
remainder are comments or built-in documentation.

### 5.5 Model and class naming scheme

The naming scheme for the classes that implement the individual terrainbento models starts with the name "Basic" and then adds a two-letter code for each element in which the model differs from the Basic model (Table 2). For example, the BasicTh model uses a threshold formula for water
erosion, but is otherwise identical to the Basic model. Model BasicRtTh uses a threshold and also implements two separate lithologies (here, "Rt" stands for "rock and till," a name that reflects the original motivation for this particular capability).

### 5.6 Input/output formats and semantics

terrainbento 1.0 provides two options for handling input of parameter values and run-control options.
Parameters can be listed in an ASCII-text input file, using YAML format ("YAML Ain't Markup Language"), as in the example in Figure 4. The name of the input file is then passed as an argument when a model object is instantiated. Alternatively, parameter name-value pairs can be entered in a Python dictionary and passed as an input when the model object is instantiated.

If a user wishes to read in a digital elevation model (DEM) to use as the initial topography, the
name of the DEM file is given as a parameter in the input file or dictionary. As of terrainbento 1.0, the file must be in ESRI ASCII format. terrainbento 1.0 treats the DEM as a watershed, using Landlab's watershed setup functionality. Any grid nodes with elevation values equal to -9999 (the ESRI "no-data" code) are set to *closed boundary* status (for more on Landlab grids, see Hobley et al., 2017). The user may optionally specify a particular grid node as the watershed outlet, using Landlab's
standard node-numbering scheme. Otherwise, an outlet node will be identified automatically. If the user does not specify the name of a DEM file, terrainbento will create a rectangular grid and initialize



its elevation field with zeros. Options to make hexagonal grids and add random noise are available and described in the User Guide. For two-lithology ("Rt") models, the user must also provide an ESRI ASCII file containing the elevations of the contact between the two units at each grid node.

Gridded output is written in netCDF format. The base name for the output files must be specified as an input parameter. When a terrainbento model runs, output is written at regular intervals, with the frequency set by the user via an input parameter. One file is created for every output interval; these files are numbered sequentially. A terrainbento output file contains all of the grid fields used in that particular model, which is to say all the grid fields created by that model's Landlab Components

plus any created in the main model program.

In addition to model output in the form of netCDF files, terrainbento supports the supply of one or more function or class, termed an "OutputWriter" that is run at output intervals. If writing and then postprocessing the netCDF files is not sufficient for a user's application, the user can define an OutputWriter to suit the application. For example, if users wanted to make a diagnostic plot to

monitor a model run as it progresses, they could define an OutputWriter that does this. The interface constraints on OutputWriters are minimal: a function must take only one argument, expected to be a terrainbento model instance; a class must take one argument at instantiation, also expected to be a model instance, and must have a "`run_one_step`" method that takes no arguments. Examples of OutputWriter usage are presented in the Jupyter Notebook "Introduction to terrainbento

OutputWriters" and in the coupled model notebooks.

Unique names are assigned to each terrainbento input parameter and each data field. terrainbento 1.0 parameter and field names are listed in Table 4, together with their equivalent mathematical symbols. terrainbento 1.0 follows the naming conventions used by Landlab (see Hobley et al., 2017). These conventions are loosely based on the CSDMS Standard Names (Peckham et al., 2013), whose

syntax uses an "object plus value" pattern (for example, *topographic__elevation*). Both Landlab and terrainbento 1.0 names seek a balance between brevity, information content, and consistency with the CSDMS Standard Names. Many of the terrainbento/Landlab names are shorter than their full Standard Name equivalents (which can be quite lengthy), but are designed to be similar enough to allow one-to-one automated mapping. Examples of input-parameter names are shown in the input file

example in Figure 4. Similar principles apply to the field names, which are encoded in the netCDF output files.





**Table 4.** terrainbento parameter names and unit dimensions.

| Symbol | Name | Dimensions |
|:---:|:---:|:---:|
| $b$ | `water_erosion_rule__thresh_depth_derivative` | $T^{-1}$ |
| $c$ | `rainfall__shape_factor` | - |
| $f$ | `climate_factor` | - |
| $h_r$ | `mean_storm_depth` | L |
| $m$ | `m_sp` | - |
| $n$ | `n_sp` | - |
| $n_{ts}$ | `number_of_sub_time_steps` | integer |
| $p_d$ | `rainfall__mean_rate` | $LT^{-1}$ |
| $D$ | `regolith_transport_parameter` | $L^2T^{-1}$ |
| $F_f$ | `fraction_fines` | - |
| $F$ | `rainfall_intermittency_factor` | - |
| $H_*$ | `roughness__length_scale` | L |
| $H_0$ | `soil_transport__decay_depth` | L |
| $H_{init}$ | `soil__initial_thickness` | L |
| $H_s$ | `soil_production__decay_depth` | L |
| $I_m$ | `infiltration_capacity` | $LT^{-1}$ |
| $K$ | `water_erodability`[2] | $T^{-1}$ |
| $K_r$ | `water_erodability rock`[2] | $T^{-1}$ |
| $K_s$ | `water_erodability sediment`[2] | $T^{-1}$ |
| $K_1$ | `water_erodability upper`[2] | $T^{-1}$ |
| $K_2$ | `water_erodability lower`[2] | $T^{-1}$ |
| $K_q$ | `water_erodability stochastic`[2] | $L^{-1/2} T^{-1/2}$ |
| $K_{sat}$ | `hydraulic_conductivity` | $LT^{-1}$) |
| $P_0$ | `soil_production__maximum_rate` | $LT^{-1}$ |
| $R_m$ | `recharge_rate` | $LT^{-1}$ |
| $S_c$ | `critical_slope` | - |
| $S_r$ | `random_seed` | integer |
| $T_b$ | `mean_interstorm_duration` | T |
| $T_r$ | `mean_storm_duration` | T |
| $T_s$ | `climate_constant_date` | T |
| $V_s$ | `normalized_settling_velocity` | - |
| $V$ | `settling_velocity` | $LT^{-1}$ |
| $W_c$ | `contact_zone__width` | L |
| $\phi$ | `sediment_porosity` | - |
| $\omega_c$ | `erosion__threshold`[1] | $LT^{-1}$ |
| $\omega_{c1}$ | `till_erosion__threshold`[1] | $LT^{-1}$ |
| $\omega_{c2}$ | `rock_erosion__threshold`[1] | $LT^{-1}$ |

[1] Becomes field rather than single-value parameter in Dd models [2] Units depend on value of $m$. Here we use $m = 0.5$.

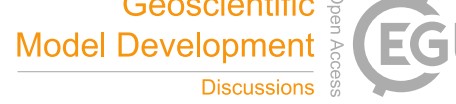

**Table 5.** terrainbento field names and unit dimensions.

| Symbol | Name | Dimensions |
|--------|------|------------|
| $A$ | drainage_area | $L^2$ |
| $A_{eff}$ | effective_drainage_area | $L^2$ |
| $H$ | soil__depth | $L$ |
| $Q$ | surface_water__discharge | $L^3T^{-1}$ |
| $Q_s$ | sediment__flux | $L^3T^{-1}$ |
| $S$ | topographic__steepest_slope | - |
| $\eta$ | topographic__elevation | $L$ |
| $\eta_b$ | bedrock__elevation | $L$ |
| $\eta_C$ | lithology_contact__elevation | $L$ |



### 5.7 Unit tests and model verification

terrainbento follows modern software engineering best practices by incorporating documentation and testing into the package source code. The terrainbento documentation Docstrings include simple examples showing, for each model, a minimal parameter dictionary. For each model, unit tests verify that all value and compatibility checks raise the correct errors and that all existing analytical solutions are reached. These unit tests ensure that any refactoring of the code, additions or improvements in later terrainbento versions, or changes to Landlab components do not change the results produced by the model. terrainbento 1.0 has 100% coverage, which means that all lines of code in the base classes, derived models, and boundary condition handlers are tested by unit or Docstring tests.

The analytical solution unit tests represent model verification. Tests of the ErosionModel base class include verification that the same random seed reproduces the same initial condition topography, that ErosionModel can work with different instantiation methods and Landlab grid types, and that ErosionModel is compatible with boundary condition handlers and OutputWriters. For base classes like the StochasticErosionModel, we test random seed reproducibility and that the sequence of rain events generated matches the desired distribution. For each model, we test at least two process end members: a case with only hillslope processes, and a case with only water erosion process. Here, for the sake of illustration, we provide the example of an analytical solution for the Basic model.

The Basic model has the following governing equation

$$\frac{\partial \eta}{\partial t} = -KA^m S^n + D\nabla^2\eta. \tag{60}$$

Given boundary conditions of a constant relative uplift rate $U$ of the core nodes, at steady state this equation becomes

$$0 = U - KA^m S^n + D\nabla^2\eta. \tag{61}$$

In the water-erosion-only endmember, $D = 0$ and the equation can be re-arranged for a relationship between slope and drainage area:

$$S = \left(\frac{U}{KA^m}\right)^{1/n}. \tag{62}$$

In unit tests we assert that the Basic model run to steady state conforms to this slope-area relationship. With regard to the hillslope-process-only case ($K = 0$), the governing equation under conditions of constant uplift can be re-arranged for an expression of elevation as a function of position within the domain. For a model domain of size $L$ in the $x$ dimension and only 1 row of core nodes in the $y$ dimension, this analytical solution is

$$\eta = \frac{U}{2D}\left(L^2 - x^2\right). \tag{63}$$

Unit tests for all other models can be found in the source code under the folder `tests`.

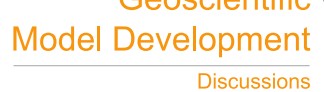



## 6 Support for users and potential developers

terrainbento includes eight Jupyter notebooks designed to introduce new users to terrainbento and demonstrate five benchmark examples. These notebooks are avaliable on GitHub at (https://github.com/terrainbento/examples_tests_and_tutorials). Three introductory notebooks go over the philosophy of terrainbento models and an introduction to using them, an overview of how to use the baselevel handler methods, and examples of creating and using OutputWriters.

The five example landscapes shown in Figure 1 are benchmark examples in which a terrainbento model is created from a Python dictionary and run to steady state with output saved to a compiled netCDF. In each of these benchmark example notebooks a 2D image of elevation and slope area plot shows example model results.

As terrainbento was designed to be generic, it includes a model template to support interested
developers to build their own model within the framework. This model template provides an example file with the skeleton of a terrainbento model and extensive comments on the type of documentation and public functions required of new terrainbento models. Throughout the documentation we have made notes encouraging users and developers to make a GitHub issue if they have questions, find errors, or feel that the functionality should be expanded to meet research needs.

## 1025 7 Conclusions

terrainbento 1.0 is a model analysis package and collection of alternative models of long-term landscape evolution built using the Landlab framework. terrainbento was designed to enable hypothesis testing among alternative models of Earth surface processes. terrainbento 1.0 focuses on 13 binary options for process formulation and 28 model programs that systematically explore these options.
As boundary conditions can substantially influence model results, terrainbento also includes five boundary condition handlers, which permit consideration of boundary conditions in a parameterized way. Integration between terrainbento and Landlab permits process component development within Landlab and use of new components in terrainbento. Thus, while the process combinations available within terrainbento 1.0 are not exhaustive, its extensible design facilitates inclusion of additional
terrainbento models using new and existing Landlab components.

Recent work has yielded a plethora of numerical models of Earth surface processes, yet comparison among models has long been difficult and inconsistent. terrainbento enables efficient model intercomparison with standardized parameters, input/output, and handling of boundary conditions. Consistent, reproducible comparison among landscape evolution models using the terrainbento mod-
eling package will support model evaluation and advance quantitative understanding of Earth surface dynamics.





## 8    Code Availability

The terrainbento source code is available in a publicly available GitHub repository distributed under a MIT license (https://github.com/terrainbento/terrainbento, https://doi.org/10.5281/zenodo.1345802),

the User Manual is built into the source code Docstrings and compiled into a Read The Docs webpage (http://terrainbento.readthedocs.io/en/latest/), and Jupyter notebooks that introduce terrainbento usage and show example models runs are located in a public GitHub repository (https://github.com/terrainbento/examples_tests_and_tutorials, https://doi.org/10.5281/zenodo.1345788). The terrainbento conda channel provides access to a conda package version of the software (author note

to reviewers and editors: as of initial manuscript submission this conda channel is not active as we anticipate incorporating any revisions into the code before distributing the package).

### 8.1    Continuous Integration and Dependencies

terrainbento 1.0 is tested with continuous integration tools TravisCI and AppVeyor to ensure that it can be installed and all tests pass on three operating systems (Windows, Ubuntu Linux, and Mac

OSX) and three Python version (2.7, 3.5, 3.6). Installing terrainbento from source requires Python and setuptools. Running terrainbento additionally requires numpy (version 1.11 or higher), scipy, xarray, dask, sympy, six, pyyaml, pytest, and Landlab (version 1.3 or higher). Testing terrainbento additionally requires pytest-cov, and pytest-runner.

### Appendix A:  Table of Mathematical Symbols





**Table 6.** Table of dimensions and constants

| Symbol | Definition | Dimensions |
|:---:|:---:|:---:|
| $x$ | x dimension | L |
| $y$ | y dimension | L |
| $z$ | z dimension | L |
| $t$ | time | T |
| $e$ | Euler's constant | - |

Table 7: Mathematical symbols[1]

| Symbol | Definition | Dimensions |
|:---:|:---:|:---:|
| $a$ | drainage area per unit contour length | L |
| $d^*$ | dimensionless number | - |
| $\mathrm{dx}_f$ | flow width | L |
| $i$ | model grid node index | - |
| $k_1$ | fluvial incision coefficient | units depend on $\mu$ |
| $p$ | precipitation rate | $LT^{-1}$ |
| $p_{ma}$ | mean precipitation rate | $LT^{-1}$ |
| $q$ | surface water unit discharge | $L^2T^{-1}$ |
| $q_{tot}$ | total unit discharge | $L^2T^{-1}$ |
| $q_s$ | fluvial sediment flux per unit width | $L^2T^{-1}$ |
| $q_h$ | hillslope sediment flux per unit width | $L^2T^{-1}$ |
| $q_{ss}$ | maximum subsurface unit discharge | $L^2T^{-1}$ |
| $r$ | runoff rate | $LT^{-1}$ |
| $w$ | lithology weight factor | - |
| $D_{50}$ | median grain size | L |
| $D_I$ | cumulative erosion | L |
| $D_s$ | deposition rate | $LT^{-1}$ |
| $E_H$ | rate of erosion by hillslope processes | $LT^{-1}$ |
| $E_i$ | instantaneous erosion rate | $LT^{-1}$ |
| $E_r$ | bedrock erosion rate | $LT^{-1}$ |
| $E_s$ | sediment entrainment rate | $LT^{-1}$ |
| $E_W$ | rate of erosion by water | $LT^{-1}$ |
| $E_{WHS}$ | rate of soil erosion | $LT^{-1}$ |
| $E_{WR}$ | rate of bedrock erosion by water | $LT^{-1}$ |





| Symbol | Definition | Dimensions |
|:---:|:---:|:---:|
| $P$ | soil production rate | $LT^{-1}$ |
| $T$ | soil transmissivity | $L^2T^{-1}$ |
| $T_{L1}$ | thickness of layer 1 | $L$ |
| $Q_{ss}$ | subsurface water discharge | $L^3T^{-1}$ |
| $Q_{tot}$ | total discharge | $L^3T^{-1}$ |
| $R$ | recharge rate | $LT^{-1}$ |
| $R_c$ | critical recharge rate | $LT^{-1}$ |
| $U$ | relative uplift rate | $LT^{-1}$ |
| $\alpha$ | saturation area scale | $L^2$ |
| $\delta_L$ | binary lithology factor | - |
| $\eta_{L2}$ | elevation of top of layer 2 | $L$ |
| $\kappa$ | climate factor rate of change | $T^{-1}$ |
| $\lambda$ | precipitation distribution scale factor | - |
| $\mu$ | generic water discharge exponent | - |
| $\nu$ | generic slope exponent | - |
| $\omega$ | erosion rate that would occur without a threshold | $LT^{-1}$ |
| $\omega_{ct}$ | erosion-depth-dependent erosion threshold | $LT^{-1}$ |
| $\rho_r$ | bedrock density | $ML^{-3}$ |
| $\rho_s$ | soil bulk density | $ML^{-3}$ |
| $\Gamma$ | gamma function | - |
| $\Delta x$ | flow width/grid cell width | $L$ |
| $\Omega_c$ | threshold under which no erosion occurs | $LT^{-1}$ |

[1] here we only list symbols not defined in Tables 4, 5, or 6.

**Appendix B: Governing equations for each terrainbento 1.0 model**

**B1 Basic**

The governing equation for elevation change in the Basic model is:

$$\frac{\partial \eta}{\partial t} = -KA^mS^n + D\nabla^2\eta, \tag{B1}$$

Parameters: $K$, $m$, $n$, and $D$.



### B2 BasicTh

BasicTh adds a threshold to the water erosion term in the Basic model:

$$\frac{\partial \eta}{\partial t} = -[\omega - \omega_c(1 - e^{-\omega/\omega_c})] + D\nabla^2 \eta, \tag{B2}$$

$$\omega = KA^m S^n. \tag{B3}$$

The threshold is smoothed such that the water erosion term approaches zero when $\omega \ll \omega_c$, and asymptotes to $\omega - \omega_c$ as $\omega \gg \omega_c$.

Parameters: $K$, $D$, $m$, $n$, and $\omega_c$.

### B3 BasicDd

BasicDd includes a threshold to the water erosion term that increases with progressive incision depth:

$$\frac{\partial \eta}{\partial t} = -[\omega - \omega_c(1 - e^{-\omega/\omega_c})] + D\nabla^2 \eta, \tag{B4}$$

$$\omega = KA^m S^n, \tag{B5}$$

$$\omega_{ct}(x,y,t) = \max(\omega_c + bD_I(x,y,t), \omega_c). \tag{B6}$$

Parameters: $K$, $m$, $n$, $D$, $b$, and $\omega_c$.

### B4 BasicHy

BasicHy uses a sediment-tracking ("hybrid") water-erosion law:

$$\frac{\partial \eta}{\partial t} = \frac{VQ_s}{A(1-\phi)} - KA^m S^n + D\nabla^2 \eta, \tag{B7}$$

$$Q_s = \int_0^s \left( [K(1-F_f)A^m S^n]_s - \left[\frac{VQ_s}{A(1-\phi)}\right]_s \right) ds. \tag{B8}$$

Parameters: $K$, $m$, $n$, $D$, $\phi$, $F_f$ and $V$.

### B5 BasicCh

BasicCh uses a nonlinear law for hillslope erosion and transport:

$$\frac{\partial \eta}{\partial t} = -KA^m S^n - \nabla q_h, \tag{B9}$$

$$q_h = -DS\left[1 + \left(\frac{S}{S_c}\right)^2 + \left(\frac{S}{S_c}\right)^4 + ... \left(\frac{S}{S_c}\right)^{2(N-1)}\right], \tag{B10}$$

where $S_c$ is a critical slope gradient. Parameters: $K$, $m$, $n$, $D$, $S_c$, $N$.

### B6 BasicSt

BasicSt uses a stochastic representation of precipitation, in which the rainfall rate $p$ is a random variable. The evolution equation is

$$\frac{\partial \eta}{\partial t} = -K\hat{Q}^m S^n + D\nabla^2 \eta. \tag{B11}$$

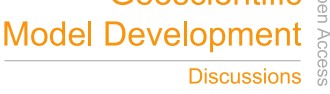



The discharge, $\hat{Q}$, associated with a particular value of $p$ is

$$\hat{Q} = p - I_m \left( 1 - e^{-p/I_m} \right), \tag{B12}$$

The probability distribution of $p$ is given by a stretched exponential survival function

$$Pr(P > p) = \exp\left[ -\left( \frac{p}{\lambda} \right)^c \right], \tag{B13}$$

with shape parameter $c$ and scale parameter $\lambda$. The relationship between $\lambda$ and the mean rainfall rate $p_d$ is

$$p_d = \lambda \Gamma(1 + 1/c). \tag{B14}$$

Parameters: $K$, $m$, $n$, $D$, $I_m$, $p_d$, $c$.

### B7 BasicVs

The BasicVs model implements variable source area runoff using the "effective area" approach described in Section 3.5.1:

$$\frac{\partial \eta}{\partial t} = -K A_{eff}^m S^n + D \nabla^2 \eta, \tag{B15}$$

$$A_{eff} = A e^{-\alpha S/A}, \tag{B16}$$

$$\alpha = \frac{K_{sat} H_{init} dx}{R_m}. \tag{B17}$$

Parameters: $K$, $m$, $n$, $D$, $K_{sat}$, $H_{init}$, $R_m$.

### B8 BasicSa

BasicSa modifies the Basic model by explicitly tracking a dynamic soil layer of thickness $H(x, y, t)$. Its governing equations are:

$$\eta = \eta_b + H, \tag{B18}$$

$$\frac{\partial H}{\partial t} = P_0 \exp(-H/H_s) - \delta(H) K A^m S^n - \nabla q_h, \tag{B19}$$

$$\frac{\partial \eta_b}{\partial t} = -P_0 \exp(-H/H_s) - (1 - \delta(H)) K A^m S^n, \tag{B20}$$

$$q_h = -D \left[ 1 - \exp\left( -\frac{H}{H_0} \right) \right] \nabla \eta. \tag{B21}$$

The function $\delta(H)$ is used to indicate that water erosion will act on soil where it exists, and on the underlying lithology where soil is absent. To achieve this, $\delta(H)$ is defined to equal 1 when $H > 0$ (meaning soil is present), and 0 if $H = 0$ (meaning the underlying parent material is exposed).

Parameters: $K$, $m$, $n$, $D$, $P_0$, $H_s$, $H_0$.



### B9 BasicRt

BasicRt modifies Basic by allowing for two lithologies, as described in Sections 2.2.2 and 3.7.2.

$$\frac{\partial \eta}{\partial t} = -K(\eta, \eta_C) A^m S^n + D\nabla^2 \eta, \tag{B22}$$

$$K(\eta, \eta_C) = wK_1 + (1-w)K_2, \tag{B23}$$

$$w = \frac{1}{1 + \exp\left(-\frac{(\eta - \eta_C)}{W_c}\right)} \tag{B24}$$

where $W_c$ is the contact-zone width.

Parameters: $K_1$, $K_2$, $m$, $n$, $D$, $W_c$ (plus specification of $\eta_C(x, y)$).

### B10 BasicCc

BasicCc uses the same governing equation as Basic, but allows the parameter $K$ to vary through time according to a linear function:

$$K(t) = \begin{cases} \mu t + fK_0, & \text{when } t < T_s, \\ K_0 & \text{otherwise.} \end{cases}, \tag{B25}$$

$$\mu = (1-f)K_0/T_s. \tag{B26}$$

Parameters: $K_0$, $m$, $n$, $D$, $f$ (factor by which $K$ is larger ($f > 1$) or smaller ($f < 1$) than $K_0$ at $t = 0$), and $T_s$ (time at which $K$ becomes constant).

### B11 BasicStTh

The land surface evolution equation is:

$$\frac{\partial \eta}{\partial t} = -\left[\hat{\omega} - \omega_c(1 - e^{-\hat{\omega}/\omega_c})\right] + D\nabla^2 \eta, \tag{B27}$$

$$\hat{\omega} = K_q \hat{Q}^m S^n. \tag{B28}$$

The discharge, $\hat{Q}$, associated with a particular value of $p$ is

$$\hat{Q} = p - I_m \left(1 - e^{-p/I_m}\right), \tag{B29}$$

The probability distribution of $p$ is given by a stretched exponential survival function

$$Pr(P > p) = \exp\left[-\left(\frac{p}{\lambda}\right)^c\right], \tag{B30}$$

with shape parameter $c$ and scale parameter $\lambda$. The relationship between $\lambda$ and the mean rainfall rate $p_d$ is

$$p_d = \lambda \Gamma(1 + 1/c). \tag{B31}$$

Parameters: $K$, $m$, $n$, $D$, $\omega_c$, $I_m$, $p_d$, $c$.



### B12 BasicThVs

The BasicThVs model implements variable source area runoff using the "effective area" approach plus a threshold on the water-erosion law:

$$\frac{\partial \eta}{\partial t} = -\left[\omega - \omega_c(1 - e^{-\omega/\omega_c})\right] + D\nabla^2\eta, \tag{B32}$$

$$\omega = KA_{eff}^{1/2}S, \tag{B33}$$

$$A_{eff} = Ae^{-\alpha S/A}, \tag{B34}$$

$$\alpha = \frac{K_{sat}H_{init}dx}{R_m}. \tag{B35}$$

Parameters: $K, m, n, D, \omega_c, K_{sat}, H_{init}, R_m$.

### B13 BasicRtTh

BasicRtTh modifies Basic by allowing for two lithologies, and applying a threshold to the channel incision law:

$$\frac{\partial \eta}{\partial t} = -\left[\omega - \omega_c(1 - e^{-\omega/\omega_c})\right] + D\nabla^2\eta, \tag{B36}$$

$$\omega = K(\eta, \eta_C)A^m S^n, \tag{B37}$$

$$K(\eta, \eta_C) = wK_1 + (1-w)K_2, \tag{B38}$$

$$\omega_c(\eta, \eta_C) = w\omega_{c1} + (1-w)\omega_{c2}, \tag{B39}$$

$$w = \frac{1}{1 + \exp\left(-\frac{(\eta - \eta_C)}{W_c}\right)} \tag{B40}$$

where $W_c$ is the contact-zone width.

Parameters: $K_1, K_2, m, n, D, \omega_{c1}, \omega_{c2}, W_c$ (plus specification of $\eta_C(x,y)$).

### B14 BasicDdHy

This is a sediment-tracking (hybrid) erosion law with a depth-dependent threshold:

$$\frac{\partial \eta}{\partial t} = \frac{VQ_s}{A(1-\phi)} - [\omega - \omega_{ct}(1 - e^{-\omega/\omega_{ct}})] + D\nabla^2\eta, \tag{B41}$$

$$Q_s = \int_0^A \left((1 - F_f)[\omega - \omega_c(1 - e^{-\omega/\omega_c})] - \frac{VQ_s}{A(1-\phi)}\right) dA, \tag{B42}$$

$$\omega = KA^m S^n, \tag{B43}$$

$$\omega_{ct}(x,y,t) = \max(\omega_c + bD_I(x,y,t), \omega_c). \tag{B44}$$

Parameters: $K, m, n, D, V, b$, and $\omega_c, \phi, F_f$.





### B15    BasicDdSt

This model uses stochastic precipitation, and the water-erosion law includes a depth-dependent
threshold:

$$\frac{\partial \eta}{\partial t} = -[\omega - \omega_{ct}(1 - e^{-\omega/\omega_{ct}})] + D\nabla^2 \eta, \tag{B45}$$

$$\omega = K_q \hat{Q}^{1/2} S, \tag{B46}$$

$$\omega_{ct}(x,y,t) = \max(\omega_c + bD_I(x,y,t), \tag{B47}$$

$$\hat{Q} = p - I_m \left(1 - e^{-p/I_m}\right), \tag{B48}$$

$$Pr(P > p) = \exp\left[-\left(\frac{p}{\lambda}\right)^c\right], \tag{B49}$$

$$p_d = \lambda\Gamma(1 + 1/c). \tag{B50}$$

Parameters: $K_q$, $m$, $n$, $D$, $I_m$, $p_d$, $c$, $\omega_c$, $b$.

### B16    BasicDdVs

Model BasicDdVs uses variable source-area hydrology, and an erosion threshold that increases with
progressive erosion depth:

$$\frac{\partial \eta}{\partial t} = -[\omega - \omega_{ct}(1 - e^{-\omega/\omega_{ct}})] + D\nabla^2 \eta, \tag{B51}$$

$$\omega = KA_{eff}^{1/2} S, \tag{B52}$$

$$A_{eff} = Ae^{-\alpha S/A}, \tag{B53}$$

$$\alpha = \frac{K_{sat}H_{init}dx}{R_m}, \tag{B54}$$

$$\omega_{ct}(x,y,t) = \max(\omega_c + bD_I(x,y,t). \tag{B55}$$

Parameters: $K$, $m$, $n$, $D$, $\omega_c$, $b$, $K_{sat}$, $H_{init}$, $R_m$.

### B17    BasicDdRt

BasicDdRt modifies Basic by allowing for two lithologies, and applying a depth-dependent threshold
to the channel incision law. Unlike BasicRtTh, the (initial) threshold is taken to be uniform across
the two lithologies; the rate of increase in threshold with depth ($b$) is also assumed uniform.

$$\frac{\partial \eta}{\partial t} = -\left[\omega - \omega_{ct}(1 - e^{-\omega/\omega_{ct}})\right] + D\nabla^2 \eta, \tag{B56}$$

$$\omega = K(\eta, \eta_C)A^m S^n, \tag{B57}$$

$$K(\eta, \eta_C) = wK_1 + (1 - w)K_2, \tag{B58}$$

$$\omega_{ct}(x,y,t) = \max(\omega_c + bD_I(x,y,t), \tag{B59}$$

$$w = \frac{1}{1 + \exp\left(-\frac{(\eta - \eta_C)}{W_c}\right)} \tag{B60}$$

where $W_c$ is the contact-zone width and $D_I(x,y,t)$ is the cumulative erosion at each point through
time.





Parameters: $K_1$, $K_2$, $m$, $n$, $D$, $\omega_c$, $b$, $W_c$ (plus specification of $\eta_C(x,y)$).

### B18 BasicHySt

$$\frac{\partial \eta}{\partial t} = \frac{VQ_s}{\hat{Q}} - K_q \hat{Q}^m S^n + D\nabla^2 \eta, \tag{B61}$$

$$Q_s = \int_0^A \left( K_q(1-F_f)\hat{Q}^m S^n - \frac{VQ_s}{A(1-\phi)} \right) dA, \tag{B62}$$

$$\hat{Q} = A\left[ p - I_m \left( 1 - e^{-p/I_m} \right) \right], \tag{B63}$$

$$Pr(P > p) = \exp\left[ -\left( \frac{p}{\lambda} \right)^c \right], \tag{B64}$$

$$p_d = \lambda \Gamma(1 + 1/c). \tag{B65}$$

Parameters: $K_q$, $m$, $n$, $V$, $D$, $I_m$, $p_d$, $c$, $\phi$, $F_f$.

### B19 BasicHyVs

Sediment-tracking (hybrid) model that uses variable source-area hydrology:

$$\frac{\partial \eta}{\partial t} = \frac{VQ_s}{A_{eff}} - KA_{eff}^m S^n + D\nabla^2 \eta, \tag{B66}$$

$$Q_s = \int_0^A \left( K(1-F_f)A_{eff}^m S^n - \frac{VQ_s}{A_{eff}} \right) dA, \tag{B67}$$

$$A_{eff} = Ae^{-\alpha S/A}, \tag{B68}$$

$$\alpha = \frac{K_{sat}H_{init}dx}{R_m}. \tag{B69}$$

Parameters: $K$, $m$, $n$, $D$, $V$, $K_{sat}$, $H_{init}$, $R_m$.

### B20 BasicHySa

This model uses a continuous layer of soil/alluvium, which influences both hillslope transport and water erosion and transport. This model configuration uses the SPACE algorithm of Shobe et al. (2017), whose governing equations can be summarized as:

$$\eta = \eta_b + H, \tag{B70}$$

$$\frac{\partial H}{\partial t} = P_0 \exp(-H/H_s) + \frac{V_s Q_s}{A(1-\phi)} - K_s A^m S^n (1 - e^{-H/H_*}) - \nabla q_h, \tag{B71}$$

$$\frac{\partial \eta_b}{\partial t} = -P_0 \exp(-H/H_s) - K_r A^m S^n e^{-H/H_*}, \tag{B72}$$

$$Q_s = \int_0^A \left( K_s A^m S^n (1 - e^{-H/H_*}) + K_r (1-F_f) A^m S^n e^{-H/H_*} - \frac{V_s Q_s}{A(1-\phi)} \right) dA, \tag{B73}$$

$$q_h = -D\left[ 1 - \exp\left( -\frac{H}{H_0} \right) \right] \nabla \eta. \tag{B74}$$

Parameters: $K_s$, $K_r$, $m$, $n$, $H_*$, $V_s$, $D$, $H_0$, $P_0$, $H_s$, $\phi$, $F_f$.



### B21 BasicHyRt

Sediment-tracking (hybrid) model with two lithologies:

$$\frac{\partial \eta}{\partial t} = \frac{VQ_s}{A(1-\phi)} - KA^m S^n + D\nabla^2 \eta, \tag{B75}$$

$$Q_s = \int_0^A \left( K(1-F_f)A^m S^n - \frac{VQ_s}{A(1-\phi)} \right) dA, \tag{B76}$$

$$K(\eta, \eta_C) = wK_1 + (1-w)K_2, \tag{B77}$$

$$w = \frac{1}{1 + \exp\left(-\frac{(\eta-\eta_C)}{W_c}\right)}. \tag{B78}$$

Parameters: $K_1$, $K_2$, $m$, $n$, $V$, $D$, $W_c$, $\phi$, $F_f$.

### B22 BasicChSa

BasicChSa modifies the Basic model by explicitly tracking a dynamic soil layer of thickness $H(x,y,t)$, and using a nonlinear (cubic) hillslope transport law. Its governing equations are:

$$\eta = \eta_b + H, \tag{B79}$$

$$\frac{\partial H}{\partial t} = P_0 \exp(-H/H_s) - \delta(H)KA^m S^n - \nabla q_h, \tag{B80}$$

$$\frac{\partial \eta_b}{\partial t} = -P_0 \exp(-H/H_s) - (1-\delta(H))KA^m S^n, \tag{B81}$$

$$q_h = -DS\left[1 + \left(\frac{S}{S_c}\right)^2 + \left(\frac{S}{S_c}\right)^4 + ... \left(\frac{S}{S_c}\right)^{2(N-1)}\right]. \tag{B82}$$

Parameters: $K$, $m$, $n$, $D$, $S_c$, $N$, $P_0$, $H_s$, $H_0$.

### B23 BasicChRt

This model uses nonlinear hillslope transport and two lithologies:

$$\frac{\partial \eta}{\partial t} = -K(\eta, \eta_C)A^m S^n - \nabla q_h, \tag{B83}$$

$$K(\eta, \eta_C) = wK_1 + (1-w)K_2, \tag{B84}$$

$$w = \frac{1}{1 + \exp\left(-\frac{(\eta-\eta_C)}{W_c}\right)}, \tag{B85}$$

$$q_h = -DS\left[1 + \left(\frac{S}{S_c}\right)^2 + \left(\frac{S}{S_c}\right)^4 + ... \left(\frac{S}{S_c}\right)^{2(N-1)}\right]. \tag{B86}$$

Parameters: $K_1$, $K_2$, $m$, $n$, $D$, $S_c$, $W_c$, $N$, (plus specification of $\eta_C(x,y)$).




### B24 BasicStVs

BasicStVs uses a stochastic representation of precipitation, together with variable source-area hydrology:

$$\frac{\partial \eta}{\partial t} = -K\hat{Q}^m S^n + D\nabla^2 \eta, \tag{B87}$$

$$\hat{Q} = pA - TS\Delta x[1 - \exp(-pA/TS\Delta x)], \tag{B88}$$

$$T = K_{sat}H, \tag{B89}$$

$$Pr(P > p) = \exp\left[-\left(\frac{p}{\lambda}\right)^c\right], \tag{B90}$$

$$p_d = \lambda\Gamma(1 + 1/c). \tag{B91}$$

Parameters: $K$, $m$, $n$, $D$, $p_d$, $c$, $K_{sat}$, and $H$ (the latter two effectively form a single lumped parameter, $T$, but each one needs to be specified in the input file).

### B25 BasicSaVs

This model combines variable source-area hydrology with a dynamic soil layer. Unlike other model configurations with variable source-area hydrology, here the actual soil thickness $H(x,y,t)$ is used to calculate transmissivity.

$$\eta = \eta_b + H, \tag{B92}$$

$$\frac{\partial H}{\partial t} = P_0\exp(-H/H_s) - \delta(H)KA_{eff}^m S^n - \nabla q_h, \tag{B93}$$

$$\frac{\partial \eta_b}{\partial t} = -P_0\exp(-H/H_s) - (1-\delta(H))KA_{eff}^m S^n, \tag{B94}$$

$$q_h = -D\left[1 - \exp\left(-\frac{H}{H_0}\right)\right]\nabla\eta, \tag{B95}$$

$$A_{eff} = A\exp\left(-\frac{-K_{sat}H\Delta xS}{R_m A}\right). \tag{B96}$$

Parameters: $K$, $m$, $n$, $K_{sat}$, $R_m$, $D$, $H_0$, $P_0$, $H_s$.

### B26 BasicRtVs

BasicRtVs is a two-lithology model configuration that uses variable source-area hydrology:

$$\frac{\partial \eta}{\partial t} = -K(\eta, \eta_C)A_{eff}^m S^n + D\nabla^2\eta, \tag{B97}$$

$$K(\eta, \eta_C) = wK_1 + (1-w)K_2, \tag{B98}$$

$$w = \frac{1}{1 + \exp\left(-\frac{(\eta - \eta_C)}{W_c}\right)}, \tag{B99}$$

$$A_{eff} = A\exp\left(-\frac{-\alpha S}{A}\right), \tag{B100}$$

$$\alpha = \frac{K_{sat}Hdx}{R_m}. \tag{B101}$$

Parameters: $K_1$, $K_2$, $m$, $n$, $K_{sat}$, $H_{init}$, $R_m$, $D$, $W_c$ (plus specification of $\eta_C(x,y)$).




### B27 BasicRtSa

This model configuration combines a dynamic soil layer and two lithologies:

$$\eta = \eta_b + H, \tag{B102}$$

$$\frac{\partial H}{\partial t} = P_0 \exp(-H/H_s) - \delta(H)KA^mS^n - \nabla q_h, \tag{B103}$$

$$\frac{\partial \eta_b}{\partial t} = -P_0 \exp(-H/H_s) - (1 - \delta(H))KA^mS^n, \tag{B104}$$

$$q_h = -D\left[1 - \exp\left(-\frac{H}{H_0}\right)\right]\nabla\eta, \tag{B105}$$

$$K(\eta, \eta_C) = wK_1 + (1 - w)K_2, \tag{B106}$$

$$w = \frac{1}{1 + \exp\left(-\frac{(\eta - \eta_C)}{W_c}\right)}. \tag{B107}$$

Parameters: $K_1$, $K_2$, $m$, $n$, $P_0$, $H_s$, $D$, $H_0$, $W_c$ (plus specification of $\eta_C(x, y)$).

### B28 BasicChRtTh

This model uses nonlinear hillslope transport, two lithologies, and an erosion threshold:

$$\frac{\partial \eta}{\partial t} = -\left[\omega - \omega_c(1 - e^{-\omega/\omega_c})\right] + D\nabla^2\eta, \tag{B108}$$

$$\omega = K(\eta, \eta_C)A^mS^n, \tag{B109}$$

$$K(\eta, \eta_C) = wK_1 + (1 - w)K_2, \tag{B110}$$

$$w = \frac{1}{1 + \exp\left(-\frac{(\eta - \eta_C)}{W_c}\right)}, \tag{B111}$$

$$q_h = -DS\left[1 + \left(\frac{S}{S_c}\right)^2 + \left(\frac{S}{S_c}\right)^4 + ... \left(\frac{S}{S_c}\right)^{2(N-1)}\right]. \tag{B112}$$

Parameters: $K_1$, $K_2$, $m$, $n$, $D$, $S_c$, $W_c$, $N$, (plus specification of $\eta_C(x, y)$).

*Acknowledgements.* We thank Eric Hutton at the Community Surface Dynamics Modeling System Integration
Facility for assistance in best practices for scientific software development and setting up continuous integration
(CSDMS is supported by NSF Award 1226297). Support for this work was provided by NSF Award 1450409 to
Tucker and an NSF EAR Postdoctoral Fellowship to Barnhart (Award Number 1725774). An early version of
1295 this software was developed in support of a project that was carried out under contract with Enviro Compliance
Solutions, Inc. (Contract Number DE-EM0002446/0920/13/DE-DT0005364/001).

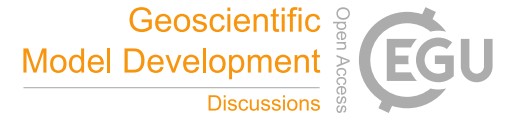



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
