# Peer review of "terrainbento 1.0: a Python package for multi-model analysis in long-term drainage basin evolution"

_Geoscientific Model Development, 2018_

## Referee Comment (RC1) · Anonymous Referee #1 · 12 Dec 2018

This paper presents a new Python modeling package for modeling Earth surface processes. With 28 model programs that result from 13 binary choices between alternate process laws, terrainbento facilitates comparison among model behavior with different process laws – a need in the geomorphic community. The new model package allows for different treatment of hillslope processes, bedrock erosion, surface water hydrology, sediment entrainment and deposition, precipitation and climate, as well as different initial and boundary conditions. terrainbento utilizes Landlab components such that incorporating new components would be relatively easy and it also incorporates model testing and verification. This package is a novel tool that will certainly advance scientific questions in Earth surface dynamics.

[Figure]

The manuscript clearly and accurately outlines all components, including all terminology used and the theory and math behind every process law, as well as model structure, organization, and implementation, such that utilizing this model package really seems like it would be straightforward. The manuscript is excellently written and very well organized – the authors have clearly demonstrated the novelty and utility in the modeling package they present. My only suggestions are two typos on line 802 and 904.

---

## Referee Comment (RC2) · Anonymous Referee #2 · 14 Jan 2019

I'll be short to avoid the system eating my contributions again.

Barnhart et al provide the geomorphic community with a very important tool: a model (in my terms) that allows easy exploration of the impacts of different choices for processes and boundary conditions.This is incredibly important and will be much appreciated. The model is well presented, very well written. I recommend it for publication with relatively minor changes. My annotated manuscript contains the smaller suggestions. Here are the larger suggestions:

1. The explanation of why 13 binary choices lead only to 28 model versions needs to move up to an earlier explanation. I (and others probably) are left wondering until too

far into the manuscript.

2. It is not well enough explained that the smoothing out of step-functions is not only for mathematical convenience (avoid daemons), but also for process- and scaling-related considerations - if I get this correctly. This generic point can be made a bit clearer the first time you present a smoothed function.

3. You provide a note on terminology in 2.1. The terms you define seem reasonable, but are then very rarely used in the ms, whereas the container term 'model' is used almost exclusively. That is not helpful. Either use the terms you define, or made 2.1 much shorter. (Or clarify what 2.1 is meant to achieve if I get it wrong).

4. The legends in Fig 1 are too small and scientific notation is not useful for these numbers.

5. I am no expert in python and code projects, so I hope you can rely on other reviewers for Chapter 5.

Again, great work, and thank you for making all this available on behalf of the community.

Please also note the supplement to this comment:
https://www.geosci-model-dev-discuss.net/gmd-2018-204/gmd-2018-204-RC2-supplement.pdf

————————————————

---

## Author Comment (AC1) · 6 Feb 2019

We thank the reviewers for their thoughtful and and constructive comments. Incorporating these comments has improved the clarity and structure of the manuscript. Based on reviewer comments we have made the following changes to the manuscript text:

- Fixed the typos pointed out by reviewer 1 in lines 802 and 904.

- In response to reviewer 2's recommendation #1 we moved the initial explanation regarding the relationship between the number of binary choices and the number of models earlier in the document. Specifically we changed text in the abstract

and in the introduction.

- We revised the text regarding numerical smoothing of functions with hard thresholds (reviewer 2's comment #2). We agree with reviewer 2 that the best place to do this is at the first mention of the smoothed functions.

- Based on reviewer 2's comment #3 on terminology, we have revised to text to qualify the word "model" wherever necessary.

- In response to reviewer 2's comment #4, we revised Figure 1 to have larger legends and no scientific notation.

- Additionally, we made many in-text changes in response to reviewer 2's supplementary file which contained a scan of the manuscript.
* * *

---

## Author Response (AR1)

February 13, 2019

Topical Editor Jackson,

Please find our response to reviewers and a marked up version of the manuscript attached to this letter. I also want to inform you of two items related to the revision of this manuscript.

- We have not yet changed the version number of the software package hosted on GitHub and distributed through Conda Forge. Thus the DOI links in the manuscript do not point to the 1.0 version. We plan to make the v1.0 release only once the manuscript has been accepted and plan to provide updated DOIs as part of final copy editing.

- Between initial submission of this manuscript in August 2018 and receipt of reviews in December 2018 and January 2018, we made a few changes to the code base (e.g. input file format changes, consistent use of discharge, instead of drainage area in governing equations) to make the 1.0 release forward compatible with the upcoming 1.0 release of the Community Surface Dynamics Modeling System Python Modeling Tool (PyMT). We have updated the manuscript to align it with the implementation changes. Accordingly, the description of model instantiation, input files, and some equations have changed.

Best Regards,

Katy Barnhart and coauthors

**Response to reviewers**

Changes made based on comments from Reviewer 1:

- Fixed the typos pointed out by reviewer 1 in lines 802 and 904.

Changes made based on comments from Reviewer 2:

- In response to reviewer 2's recommendation #1 we moved the initial explanation regarding the relationship between the number of binary choices and the number of models earlier in the document. Specifically we changed text in the abstract and in the introduction.

- We revised the text regarding numerical smoothing of functions with hard thresholds (reviewer 2's comment #2). We agree with reviewer 2 that the best place to do this is at the first mention of the smoothed functions.

- Based on reviewer 2's comment #3 on terminology, we have revised to text to qualify the word "model" wherever necessary.

- In response to reviewer 2's comment #4, we revised Figure 1 to have larger legends and no scientific notation.

- Additionally, we made many in-text changes in response to reviewer 2's supplementary file which contained a scan of the manuscript.

[revised manuscript text omitted]

[1] here we only list symbols not defined in Tables 4, 5, or 6.

**Appendix B: Governing equations for each terrainbento 1.0 model program**

**B1 Basic**

The governing equation for elevation change in the Basic model  program is:

$$\frac{\partial \eta}{\partial t} = -\underline{KA}KQ^m S^n + D\nabla^2 \eta, \tag{B1}$$

Parameters: $K$, $m$, $n$, and $D$.

**B2   BasicTh**

1135  BasicTh adds a threshold to the water erosion term in the Basic model  program:

$$\frac{\partial \eta}{\partial t} = -[\omega - \omega_c(1 - e^{-\omega/\omega_c})] + D\nabla^2\eta \,, \tag{B2}$$

$$\omega = K\underline{A}Q^m S^n \,. \tag{B3}$$

The threshold is smoothed such that the water erosion term approaches zero when $\omega \ll \omega_c$, and asymptotes to $\omega - \omega_c$ as $\omega \gg \omega_c$.

1140  Parameters: $K$, $D$, $m$, $n$, and $\omega_c$.

**B3   BasicDd**

BasicDd includes a threshold to the water erosion term that increases with progressive incision depth:

$$\frac{\partial \eta}{\partial t} = -[\omega - \omega_c(1 - e^{-\omega/\omega_c})] + D\nabla^2\eta \,, \tag{B4}$$

$$\omega = K\underline{A}Q^m S^n \,, \tag{B5}$$

1145  $$\omega_{ct}(x, y, t) = \max(\omega_c + bD_I(x, y, t), \omega_c) \,. \tag{B6}$$

Parameters: $K$, $m$, $n$, $D$, $b$, and $\omega_c$.

**B4   BasicHy**

BasicHy uses a sediment-tracking ("hybrid") water-erosion law:

$$\frac{\partial \eta}{\partial t} = \frac{VQ_s}{A(1 - \phi)} - \underline{KA}KQ^m S^n + D\nabla^2\eta \,, \tag{B7}$$

1150  $$Q_s = \int_0^{\underline{s}\underline{A}} \left( \left[ K(1 - F_f)\underline{A}Q^m S^n \right]_s - \left[ \frac{VQ_s}{A(1 - \phi)} \
[revised manuscript text omitted]

---

## Author Response (AR2)

February 15, 2019

Topical Editor Jackson and GMD staff,

Please find our response to reviewers and a marked up version of the manuscript attached to this letter.
In my initial revised submission I informed Topical Editor Jackson of the following to items related to the revision of this manuscript.

- We have not yet changed the version number of the software package hosted on GitHub and distributed through Conda Forge. Thus the DOI links in the manuscript do not point to the 1.0 version. We plan to make the v1.0 release only once the manuscript has been accepted and plan to provide updated DOIs as part of final copy editing.

- Between initial submission of this manuscript in August 2018 and receipt of reviews in December 2018 and January 2018, we made a few changes to the code base (e.g. input file format changes, consistent use of discharge, instead of drainage area in governing equations) to make the 1.0 release forward compatible with the upcoming 1.0 release of the Community Surface Dynamics Modeling System Python Modeling Tool (PyMT). We have updated the manuscript to align it with the implementation changes. Accordingly, the description of model instantiation, input files, and some equations have changed.

In response to Editor Jackson's technical correction instructions we have made the v1.0 version release on GitHub and conda-forge, created new Zenodo DOIs and updated the DOIs in the manuscript. Editor Jackson's comments also indicated that we should attend to the minor comments of reviewer 1. As noted in the previous response to reviewers and the public author comment, we have already addressed these comments. We also added one sentence to the acknowledgments section of the manuscript.
Best Regards,

Katy Barnhart and coauthors

**Response to reviewers**

Changes made based on comments from Reviewer 1:

- Fixed the typos pointed out by reviewer 1 in lines 802 and 904.

Changes made based on comments from Reviewer 2:

- In response to reviewer 2's recommendation #1 we moved the initial explanation regarding the relationship between the number of binary choices and the number of models earlier in the document. Specifically we changed text in the abstract and in the introduction.

- We revised the text regarding numerical smoothing of functions with hard thresholds (reviewer 2's comment #2). We agree with reviewer 2 that the best place to do this is at the first mention of the smoothed functions.

- Based on reviewer 2's comment #3 on terminology, we have revised to text to qualify the word "model" wherever necessary.

- In response to reviewer 2's comment #4, we revised Figure 1 to have larger legends and no scientific notation.

- Additionally, we made many in-text changes in response to reviewer 2's supplementary file which contained a scan of the manuscript.

[revised manuscript text omitted]
/001). We thank members of the Erosion Working Group for their collaboration and insights into the original application that motivated this software.

[revised manuscript text omitted]